# Effects of Yogurt with Carao (*Cassia grandis*) on Intestinal Barrier Dysfunction, α-glycosidase Activity, Lipase Activity, Hypoglycemic Effect, and Antioxidant Activity

**Ricardo S. Aleman [1], Jhunior Marcia [2], Ryan Page [1], Shirin Kazemzadeh Pournaki [3], Daniel Martín-Vertedor [4], Víctor Manrique-Fernández [5], Ismael Montero-Fernández [5] and Kayanush Aryana [1,\*]**

[1] School of Nutrition and Food Sciences, Louisiana State University Agricultural Center, Baton Rouge, LA 70802, USA; rsantosaleman@lsu.edu (R.S.A.); rpage1@lsu.edu (R.P.)

[2] Faculty of Technological Sciences, Universidad Nacional de Agricultura Road to Dulce Nombre de Culmi, Km 215, Barrio El Espino, Catacamas 16201, Honduras; juniorabrahamm@yahoo.com

[3] Department of Dairy and Food Science, South Dakota State University, Brookings, SD 57007, USA; shirin.kazemzadehpournaki@sdstate.edu

[4] Technological Institute of Food and Agriculture CICYTEX-INTAEX, Junta of Extremadura, Avda. Adolfo Suárez s/n, 06007 Badajoz, Spain; daniel.martin@juntaex.es

[5] Área de Nutrición y Bromatología, Departamento de Producción Animal y Ciencia de los Alimentos, Escuela de Ingenierías Agrarias, Universidad de Extremadura, 06006 Badajoz, Spain; vicmanfer@gmail.com (V.M.-F.)

\* Correspondence: karyana@agcenter.lsu.edu

**Abstract:** Cell inflammation disrupts intestinal barrier functions and may cause disorders related to a leaky gut, possibly leading to diabetes. The objective of this study was to determine if carao (*Cassia grandis*) incorporated into yogurt enhances in vitro intestinal barrier function. To achieve this goal, Caco-2 cells were used as a model of intestinal barrier permeability. Caco-2 cells were treated with cytokines (interleukin-1β, tumor necrosis factor-α, interferon-γ, and lipopolysaccharide (LPS)) and yogurt with carao yogurt (CY) at different doses (1.3 g/L, 2.65 g/L, and 5.3 g/L). Real-time quantitative polymerase chain and immunofluorescence microscopy were applied to evaluate the expression and localization of tight junction proteins. Functional effects of the formulation of yogurt supplemented with carao were also evaluated in terms of the antioxidant activity, the α-glycosidase activity, and lipase inhibitory properties. In addition, the hypoglycemic potential was validated in vivo in a rat model. Compared to the control yogurt, Caco-2 TEER (transepithelial electrical resistance evaluation) by yogurt with 5.3 g/L of carao was significantly lower ($p < 0.05$) after 48 h. Yogurt with 5.3 g/L of carao had a considerably lower permeability ($p < 0.05$) than control yogurt in FD and LY flux. Yogurt with 5.3 g/L of carao enhanced the localization of ZO-1. Carao addition into yogurt increased the flavonoid content, apparent viscosity, lipase inhibition activity, and α-glycosidase activity. The rats fed with the yogurt with 5.3 g/L of carao demonstrated a higher blood glucose modulation.

**Keywords:** yogurt; carao; inflammatory properties; antioxidant α-glycosidase; diabetes

## 1. Introduction

The cause of leaky gut syndrome is not clear. The digestive system is susceptible to stimuli in many people with this disorder. Affected people feel discomfort caused by intestinal gas or contractions that others would not find problematic. For some people, high-calorie foods or high-fat diets may be the trigger. Since many food products contain multiple ingredients, it is difficult to identify the specific precipitating factor.

How the precipitating factors cause the syndrome is not evident. Emotional factors (stress, anxiety, depression, and fear), drugs (including laxatives), or hormones can trigger or worsen the leaky gut syndrome. Affected people do not always have symptoms after a common trigger, and symptoms often appear without an obvious trigger [1]. On the other

hand, diabetes is a long-lasting health condition that is originated from an imbalance in the blood carbohydrates metabolism and is also related to leaky gut disorders. The World Health Organization stated that diabetes mellitus (DM) is a world epidemy [2] since it leads to blindness, kidney failure, heart attacks, stroke, and lower limb amputation. In 2016, about 1.6 million deaths were mainly driven by diabetes [3]. In January 2021, the CDC reported that out of the US population of 37.3 million people, 23% had diabetes [4]. Diabetes mellitus is a metabolic disease caused by increased blood glucose with a forecast of more than 400 million people by the year 2050 [5]. At least 537 million patients aged 20–79 (10.5%) have been diagnosed with diabetes globally [6].

The cause of diabetes, which is related to leaky gut, is unknown and complex, but the most likely cause is that this disease is led by multiple factors related to metabolic deregulation. Therefore, examinations for determining new food products for assisting DM treatments or leaky gut disorders are exceptionally crucial, and the development of this type of functional food should be carried out through a comprehensive and integrative approach. DM may cause an increase in the permeability of the gut barrier [7], leading to systemic inflammation and thereby driving metabolic disorders such as obesity, hypertension, atherosclerosis, and others [8]. For glycemic control, the best treatments found thus far have been periodontal treatments that help restore insulin sensitivity over time [9].

The fortification of yogurt with functional ingredients can improve the functionality of yogurt. Probiotics, polyphenols, lipids, and proteins have been reported to enhance intestinal barrier function by inhibiting inflammation [10–13]. In addition, low-fat yogurt can inhibit intestinal barrier dysfunction by increasing tight junctions [14]. Authors such as El-Dein et al. (2022) [15] determined that *Lactobacillus*-fermented yogurt exerts hypoglycemic, hypocholesterolemic, and anti-inflammatory activities in STZ-induced diabetic Wistar rats.

Plant extracts have been reported to be a great source of phenolics, flavonoids, terpenes, and other secondary metabolites, which could contribute to reducing the blood glucose level [16–18]. The *C. grandis* (carao) has been recommended for nutritional and pharmacological application [19,20]. For centuries, indigenous groups from Latin American countries such as Honduras, Costa Rica, Brazil, Colombia, Mexico, and Cuba have used *C. grandis* (*Cesalpinaceae*) to treat anemia, skin ulcers, and diabetes [21,22]. The pulp reports high amounts of carotenoids, whereas the seeds have high concentrations of phenolic compounds and antioxidant activity [23]. The fruit extract of *C. grandis* holds flavonoids, phenols, coumarins, saponins, and triterpenes [20]. The fruit extract of *C. grandis* reports an inhibitory effect of α-glucosidase and pancreatic lipase higher than Acarbose[®] and Orlistat[®]. Furthermore, the fruit extract of *C. grandis* reported a hypoglycemic effect in rats comparable to glibenclamide [24].

We were interested to examine if yogurt with carao can directly affect intestinal barrier function, antioxidant activity, α-glycosidase and lipase inhibitory effect, and hypoglycemic effect, and examine the functional and microbial properties of this yogurt. As a result, the purpose of this study was to examine the effects of yogurt with carao on intestinal barrier dysfunctions in Caco-2 cells and the hypoglycemic effect in an animal model.

## 2. Materials and Methods

### 2.1. Plant Material

The fruit of carao (*C. grandis*) was collected from Catacamas Municipality, Olancho Department, Honduras. The pilled pulp was ground by using a 501–700 mm knife mill (Retsch GmbH, SM 100, Haan, Germany). Then, the grounded pulp was mixed with water (10% *wt/wt*) using a magnetic stirrer at 500 rpm (Bimarloga Científica, 78HW-1, Buenos Aires, Argentina) at 55 °C for 30 min. The mixture was freeze-dried (Virtis Advantage Pro, SP Scientific, Warminster, PA, USA) and stored for further use.

## 2.2. Yogurt Preparation

Yogurt was produced as that of Aleman et al.'s work (2023) [25]. A total of 2% fat bovine milk obtained from a local supermarket was mixed with carao powder at different concentrations (0 g/L (control), 1.3 g/L, 2.65 g/L, and 5.3 g/L) and kept under constant stirring for 1 h at 10–15 °C. Subsequently, the milk was pasteurized at 85 °C for 30 min and, carao was then mixed vigorously with a commercial immersion blender (Waring Commercial, McConnellsburg, PA, USA) for 3 to 5 min while the temperature was held at 80 °C. Later, 3 mL of each culture of *S. thermophilus* ST-M5 and *L. bulgaricus* LB-12 (Chr. Hansen, Milwaukee, WI, USA) were incorporated into the milk (3 gallons) at 41 °C, and the milk mixture was then incubated at 40 ± 1 °C and kept until the fermentation pH reached 4.6. The mixture was poured into labeled 355 mL Reynolds RDC212-Del-Pak Combo-Pak containers (Alcoa, Inc., Pittsburgh, PA, USA) and stored at 4 °C.

## 2.3. Technological and Microbial Analysis

The viscosity, titratable acidity (TA), pH, and microbial counts were determined at 0, 2.5, 5, and 7 h during incubation, respectively. The apparent viscosity of yogurt was evaluated by using Brookfield rotational DV-II viscometer (Brookfield Engineering Lab Inc., Stoughton, MA, USA) on a helipad stand and using a T-C spindle rotated at 20 rpm. The pH was measured by using a pH meter (Thermo Orion 3-Star Benchtop pH Meter, Fisher Scientific, Pittsburgh, PA, USA). TA was achieved by using 9 g of yogurt with 9 mL of distilled water and 0.5 mL of phenolphthalein, and samples were titrated with 0.1 N NaOH [26]. Yogurts were serially diluted in sterilized peptone (0.1% *wt/v*) from $10^{-1}$ to $10^{-6}$ and pour plates in duplicate recording were incubated [27] *S. thermophilus* was enumerated by using *S. thermophilus* (ST) agar (formulated with sucrose, K2HPO4, Bacto yeast extract, Bacto tryptone and agar) adjusted to pH 6.8 and incubated at 37 °C for 48 h aerobically, and *L. bulgaricus* (LB) was enumerated by using *Lactobacillus* MRS agar (formulated with MRS broth powder and agar), adjusted to pH 5.2 and incubated at 43 °C for 72 h anaerobically. Colonies were counted by using a Quebec Darkfield colony counter (Leica Inc., Buffalo, NY, USA). The counts were transformed as logarithms of the number of colony-forming units per mL Log (CFU/mL). All experiments were conducted in triplicates.

## 2.4. Antioxidant Determination, Total Phenolics, and Flavonoid Content

The yogurt was first frozen to −20 °C, and then placed in a FreeZone (Labconco) freeze dryer unit for ~72 h. The freeze-dried yogurt was then pulverized using a knife mill Retsch SM 100 (Retsch GmbH, Haan, Germany) (501–700 mm). Later, 200 mg of ground lyophilized yogurt was placed in a 30 mL centrifuge falcon tube with 20 mL of 80% methanol solution. The obtained mixture was mixed and sonicated for 20 min. The sonicated solution was then centrifuged for 7 min at $1500\times g$ force. The supernatant was filtered for antioxidant determination, total phenolics, and flavonoid content.

2,2-Diphenyl-2-picrylhydrazyl (DPPH) inhibition was determined [28]: in brief, 16 µL extract of yogurt samples (YS) was mixed with 320 µL of 0.15 mM DPPH solution (Sigma-Aldrich, Schnelldorf, Germany) and 16 µL of water, which served as control. The obtained solution was stirred and incubated for 30 min at 25 °C. The absorbance was recorded (515 nm) with an ELISA reader (Synergy 2; BioTek Instruments, Winooski, VT, USA). The DPPH inhibition (%) was estimated by using Equation (1):

$$\% \text{ Inhibition} = \left( \frac{Absorbance\ control - Absorbance\ sample}{Absorbance\ control} \right) \times 100 \qquad (1)$$

Total phenolic content (TPC) was evaluated with slight modifications [29]. An amount of 100 µL extract of yogurt samples (YS) was mixed thoroughly for 3 min at 25 °C with 100 µL of 1 N Folin–Ciocalteu reagent (Sigma-Aldrich, Taufkirchen, Germany). The obtained solution was mixed with $Na_2CO_3$ (200 µL, 1 N) and held still for 90 min. The

absorbance (725 nm) was recorded with an ELISA reader (Synergy 2; BioTek Instruments, Winooski, VT, USA). The TPC is expressed as gallic acid equivalent.

Total flavonoid content (TFC) was evaluated with slight modifications [29]. An amount of 100 μL extract of yogurt samples (YS) was mixed with diethylene glycol (500 μL) and NaOH (50 μL, 1 N). The obtained solution was incubated for 60 min at 37 °C. The absorbance (420 nm) was recorded with an ELISA reader (Synergy 2; BioTek Instruments, Winooski, VT, USA). The TFC is expressed as quercetin equivalent (Sigma-Aldrich, Taufkirchen, Germany). All experiments were performed in triplicates.

### 2.5. Enzymatic Inhibition Assays

The α-glucosidase inhibitory activity was applied according to the reference method with slight modification [30]. Yogurt with carao at different concentrations (0 g/L (control), 1.3 g/L, 2.65 g/L, and 5.3 g/L) were tested. In 96-microwell plates, 50 μL of yogurt samples (YS) contained 0.5 mg/mL for YS solution and α-glucosidase from Saccharomyces cerevisiae (100 μL) (0.6 U/mL) in 0.1 M phosphate buffer pH 6.9. The obtained solution was incubated for 10 min at 37 °C. An amount of 50 μL of 3 mM p-nitrophenyl-α-glucopyranoside (NPG) in phosphate buffer pH 6.9 (NPG) was then mixed and incubated again for 13 min at 37 °C. The α-glucosidase inhibitory activity was evaluated using a spectrophotometer (Synergy HT, Bio-Tek Instruments, Inc., Winooski, VT, USA) at a wavelength of 405 nm. An amount of 1 mM acarbose (Sigma-Aldrich, Taufkirchen, Germany) was used as a control. The α-glucosidase activity (I) was calculated using Equation (2):

$$I = 100 - (AS/AC) \times 100 \qquad (2)$$

where *AS* is the difference between the absorbance of the sample and the absorbance of the blank and *AC* is the difference between the absorbance of the control and the absorbance of the blank.

The lipase inhibitory activity was observed by measuring the production of p-nitrophenol (pNP), resulting from the hydrolyzation of p-nitrophenol palmitate (pNPP) [30]. In 96-microwell plates, 20 μL YS comprising 0.5 mg/mL yogurt solution and 20 μL lipase (from the porcine pancreas) solution (1 mg/mL) in 0.1 M sodium phosphate buffer adjusted to pH 6.9 was incubated for 11 min at 37 °C. Then, 1800 μL of sodium phosphate buffer 0.1 M comprising 20 μL of p-nitrophenyl palmitate in isopropanol (0.01 M), Arabic gum (0.55 mg/mL), and sodium cholate (1.15 mg/mL) were combined and incubated at 37 °C for 11 min. The content of p-nitrophenol was measured by using a spectrophotometer (Synergy HT, Bio-Tek Instruments, Inc., Winooski, VT, USA). The lipase inhibitory activity was calculated by Equation (2). All experiments were performed in triplicates.

### 2.6. Caco-2 Cell Culture Maintenance and an Induction of Barrier Dysfunction

Caco-2 cells (human intestinal epithelial cell line) were purchased from the American Type Culture Collection (ATCC® HTB-37™), and the cells were treated in high glucose DMEM (Dulbecco's modified Eagle medium), controlled by using 10% fetal bovine serum (FBS; Thermo Fisher Scientific, Waltham, MA, USA) and 1% non-essential amino acid solution (NEAA) (*v/v*), 50 μM thioglycerol, and 25 mg/mL gentamycin. The medium was changed every 2–3 days. Caco-2 cells were seeded onto polycarbonate membrane Transwell (Corning, Inc; Lowell, MA, USA) inserts with 0.4 μm pore size at $2 \times 10^5$ cells/mL for 3–4 days to reach 80–90% confluence. To reach differentiation, cells were cultured for 21 days. The cells were incubated at 37 °C in a humidified atmosphere containing 5% $CO_2$. At wanted confluence (80–90%), cells were detached with trypsin [31,32].

Differentiated Caco-2 monolayers were treated with an inflammatory stimulus such as interleukin-1β (IL-1β), tumor necrosis factor-α (TNF-α), interferon-gamma (IFN-γ), lipopolysaccharide (LPS), and isoflavone genistein. In growth media, 25 ng·mL$^{-1}$ of IL-1β, 50 ng·mL$^{-1}$ of TNF-α, and 50 ng·mL$^{-1}$ of IFN-γ were used in the basolateral compartment, whereas LPS was applied to both the apical and basolateral compartments at 1 μg mL$^{-1}$ [33]. In addition, the cells were treated with isoflavone genistein (positive control). For yogurt

treatments, all yogurt samples with and without carao (0 g/L (control), 1.3 g/L, 2.65 g/L, and 5.3 g/L) were lyophilized as described above and diluted (1:8 (*w/v*) on a wet weight basis) in the culture media. All experiments were conducted in triplicates.

### 2.7. Transepithelial Electrical Resistance

Cell monolayer integrity influenced by the yogurt samples with and without carao (0 g/L (control), 1.3 g/L, 2.65 g/L, and 5.3 g/L) were evaluated by using transepithelial electrical resistance evaluation (TEER) [34]. The inflammatory stimulus (I) containing interleukin-1β (IL-1β), tumor necrosis factor-α (TNF-α), interferon-gamma (IFN-γ), and lipopolysaccharide (LPS) was applied with yogurt samples, and I with 100 μM genistein media (positive control) was also evaluated. Growth media were transferred from culture plates and inserts, and cells were washed with Hank's balanced salt solution (HBSS) containing 5 mM HEPES combined with the basolateral and apical compartments in different volumes (1 and 0.2 mL, respectively). Later, the plates were kept in 5% $CO_2$, at 37 °C, and for 25–30 min. Plates were then transferred to a hot plate at 39 °C and TEER was estimated by using an Epithelial Volt/Ohm (TEER) Meter 3 with an EVOM3 chopstick electrode (World Precision Instruments, Sarasota, FL, USA). An electrode was kept under the standard condition of not touching the monolayer. Samples without cells were used as blank and differentiated monolayers. TEER ($\Omega \cdot cm^2$) values were from 550–700 ohm $cm^2$ ($n = 25$). Triplicated measurements were taken. All experiments were performed in triplicates.

### 2.8. Paracellular Permeability

Paracellular permeability as influenced by the yogurt samples with and without carao (0 g/L (control), 1.3 g/L, 2.65 g/L, and 5.3 g/L) were evaluated by applying the flux of fluorescein isothiocyanate (FITC)-dextran 4000 (FD) and Lucifer yellow (LY) through the cell monolayers based on a literature review [35,36]. The inflammatory stimulus (I) containing interleukin-1β (IL-1β), tumor necrosis factor-α (TNF-α), interferon-gamma (IFN-γ), and lipopolysaccharide (LPS) was applied with yogurt samples, and I with 100 μM genistein media (positive control) was also evaluated. FD (1 mg mL$^{-1}$) and LY (0.5 mg mL$^{-1}$) were mixed with HBSS/HEPES at 37 °C and 0.2 mL of it was added into the apical compartment, whereas 1.0 mL of HBSS was added to the basolateral well. The covered plates were incubated at 37 °C and 150 rpm. An amount of 300 μL of the basolateral chamber was transferred into a black, clear-bottom 96-well plate (Corning Costar, New York, NY, USA) every 4 h for 16 h. Later, the basolateral compartment was rinsed and a new 1 mL of HBSS was added at 37 °C. The fluorescence strength was measured by using a fluorescence spectrophotometer PLX800 (BioTek, Winooski, VT, USA) under excitation/emission wavelengths of 485/530 nm (FD) and 428/540 nm (LY). The apparent permeability coefficient, $P_{app}$ (cm/s), was measured according to Equation (3):

$$P_{app}\left(cm\ s^{-1}\right) = \frac{dQ}{dt} \times \frac{1}{A \times C} \tag{3}$$

where *C* is the initial concentration on the apical side (mol/mL), *dQ* is the portion of the fluorescent marker on the basolateral side (mol/mL), *dt* is the flux per second (1/s), and *A* is the membrane surface area ($cm^2$). All experiments were performed in triplicates.

### 2.9. Immunofluorescence Microscopy

The immunofluorescence microscopy as influenced by the yogurt samples with and without carao (0 g/L (control), 1.3 g/L, 2.65 g/L, and 5.3 g/L) were evaluated [34]. The inflammatory stimulus (I) containing interleukin-1β (IL-1β), tumor necrosis factor-α (TNF-α), interferon-gamma (IFN-γ), and lipopolysaccharide (LPS) was applied with yogurt samples and cells were also treated with only the growth media (healthy cells). The cells were grown in the Lab-Tek II chamber slide system (Nalge Nunc International, Rochester, NY, USA) and with PBS (without Ca and Mg) and fixed with 3% paraformaldehyde. Cells were overlaid

with primary antibody claudin-1, occluding, and Zo-1 (Zymed Laboratories, San Francisco, CA, USA), which were diluted 1:50 and incubated for 1 h. Following incubation, secondary antibodies 1:100 (goat anti-rabbit IgG antibody, (H + L) FTIC conjugate; Sigma) were added to fixed cells and incubated for 30 min. Next, tight junction protein expression was visualized using a fluorescent light microscope with FITC-compatible media (Compound Microscope Leitz Optilux, Stuttgart, Germany). Images were recorded and examined using ZEN 2010 software (Carl Zeiss AG, Oberkochen, Germany). At least ten fields were viewed and selected based on balanced staining, and the square cutout borders were withdrawn.

### 2.10. Transmission Electron Microscopy

The transmission electron microscopy (TEM) influenced by the yogurt samples with and without carao (0 g/L (control), 1.3 g/L, 2.65 g/L, and 5.3 g/L) were evaluated [37]. The inflammatory stimulus (I) containing interleukin-1β (IL-1β), tumor necrosis factor-$\alpha$ (TNF-$\alpha$), interferon-gamma (IFN-$\gamma$), and lipopolysaccharide (LPS) was applied with yogurt samples and cells were also treated with only the growth media (healthy cells). An amount of 1 μm of cells in different treatments were collected by applying glass knives on an ultramicrotome (Reichert Ultracut S, Minnesota, MN, USA) after dehydration and inserting in Araldite epoxy resin and colored with 1% toluidine blue in 1% sodium borate (Sigma-Aldrich, Taufkirchen, Germany). The areas which were selected ($\sim$90 nm in thickness) for examination were cut by applying a diamond knife and colored with saturated uranyl acetate and Reynold's lead citrate (Sigma-Aldrich, Taufkirchen, Germany). Digital images (from 10 different clear spots) were collected from Samples that were observed by using TEM (Hitachi H7000, 100 kV, Yokohama, Japan). At least three images from each treatment group were analyzed. All experiments were conducted in triplicates.

### 2.11. Gene Expression Analysis of Tight Junction Proteins

The gene expression analysis of tight junction proteins influenced by the yogurt samples with and without carao (0 g/L (control), 1.3 g/L, 2.65 g/L, and 5.3 g/L) were evaluated [38]. The inflammatory stimulus (I) containing interleukin-1β (IL-1β), tumor necrosis factor-$\alpha$ (TNF-$\alpha$), interferon-gamma (IFN-$\gamma$), and lipopolysaccharide (LPS) was applied with yogurt samples and cells were also treated with only the growth media (healthy cells). RNA extraction from the Caco-2 cells method was applied by using a slight modification [34]. According to the methodology, $5 \times 10^6$ cells were collected using a centrifuge, and DNaseI from Ambion DNA-free$^{TM}$ Kit (Thermos Fisher Scientific, Waltham, MA, USA) was added to them for cleaning DNA residuals. RevertAid RT kits were applied to observe reversed transcription according to protocol. Qubit (Thermo Fisher Scientific, Waltham, MA, USA) was applied to define RNA concentration by using fluorometric quantitation. A total of 1 μg of isolated RNA as control was selected, and synthesized cDNA was amplified in 7500 real-time PCR using the RevertAid First Strand cDNA Synthesis Kit (Thermo Scientific, Waltham, MA, USA).

Real-time quantitative polymerase chain reaction (RT-qPCR) was achieved using iTaq™ Universal SYBR-Green Supermix on a Bio-Rad CFX96 system. The NCBI primer design tool was used to design primers for ZO-1, claudin-1, occludin, ribosomal protein large P0, and 18sRNA (RNA18S5) [38,39]. Gene expression was standardized to the mathematical mean of RPLP0 and RNA18S5, as previously validated in Caco-2 cells [40,41]. All experiments were performed in triplicates.

### 2.12. Animal Models and Glucose Tolerance Evaluation

The established animal experiment guidelines by the Declaration of Helsinki were used as a reference for the animal models, which is approved by the Ethics Committee in Food Research at the Honduran Association of Medicine and Nutrition (ASOHMENU) with form number AS-ASHOMENU-0013-2022.

Forty-nine healthy five-week-old male Wistar rats were purchased from Universidad Nacional Autonoma de Honduras, Tegucigalpa (Honduras). Rats were divided into



7 groups, fed with YS (I–IV) (carao with 0 g/L (I), 1.3 g/L (II), 2.65 g/L (III), 5.3 g/L (IV)), regular diet only (healthy or untreated rats) (V), sucrose solution (diabetic induced) (VI), and metformin (VII). First, the rats were subjected to a diabetes induction by intraperitoneal injection with streptozotocin after one week of adoption at 22 °C, 55% RH, and a regular diet (Ban et al., 2020) [42]. Injection of streptozotocin (streptozotocin freshly dissolved in 50 mmol/L citrate buffer (pH 4.5)) was administered until blood sugar levels were greater than 10.0 mmol/L, which is categorized as diabetes. After induction of diabetes, the rats were fed with highly contained sugar food (30 mg/mL sucrose at the dose of 0.1 g per 100 g body weight given). Diabetic-induced rats received a high-fat diet. Glucose levels were recorded weekly for 6 weeks after feeding. Fasting blood glucose was observed by using a OneTouch Ultra blood glucose meter (Accu-Chek Roche, Mannheim, Germany) after 12 h of fasting every two weeks, using blood collected from the tail vein. The glucose tolerance was observed by an oral glucose tolerance test (Chen et al., 2016) [43]. After 12 h of fasting, rats were injected with 2.5 g/kg glucose. Then, blood sugar samples were collected from the tail vein at 0-, 30-, 60-, and 120-min. The area under the curve (AUC) was calculated using the following Equation (4):

$$AUC = 1/4 \times A\ (0\ min) + 1/2 \times A\ (30\ min) + 3/4 \times A\ (60\ min) + 1/2 \times A\ (120\ min) \quad (4)$$

where A is the blood sugar.

### 2.13. Statistical Analysis

The statistical analysis was performed as in Aleman et al.'s work (2023) [25]. Proc mixed was used to determine the statistical significance of the carao concentration effect, hour effect, and interaction effect of carao concentration x hour for the technological measurements, microbial evaluations, transepithelial electrical resistance determinations, paracellular permeability observations, and blood glucose levels. In our case study, we used mixed proc model random (replications in yogurt batches) and mixed effect (treatments; carao concentrations), and repeated measures (time). The differences of least square means were used to determine significant differences at $p < 0.05$. In addition, one-way ANOVA was applied for total phenolics, total flavonoids, antioxidant capacity measurements, gene expression analysis, AUC measurements, fasting blood glucose levels, TEER values, enzymatic activity, and viscosity observations. Significant differences were determined at $\alpha = 0.05$. Post hoc tests were conducted with Tukey's HSD. Data were processed using Statistical Analysis Systems SAS (SAS Institute Inc., Cary, NC, USA). Each experiment was carried out in triplicate or otherwise defined.

## 3. Results and Discussion

### 3.1. Technological and Microbial Evaluation

TA values of the yogurt are illustrated in Figure 1a. The main effects of the concentration and hour were significant ($p < 0.05$), whereas the interaction effect concentration * hour was not significant ($p > 0.05$). Yogurts with 5.3 g/L of carao had significantly ($p < 0.05$) higher TA values than control samples. TA remained stable until 2.5 h and significantly ($p < 0.05$) increased after 5 h during the fermentation.

The pH values of YS during the fermentation process is illustrated in Figure 1b. The time effect was significant ($p < 0.05$), whereas the concentration effect and the concentration * time interaction effect were not significant ($p > 0.05$). pH values decreased significantly ($p < 0.05$) during 0, 2.5, 5, and 7 h of fermentation.

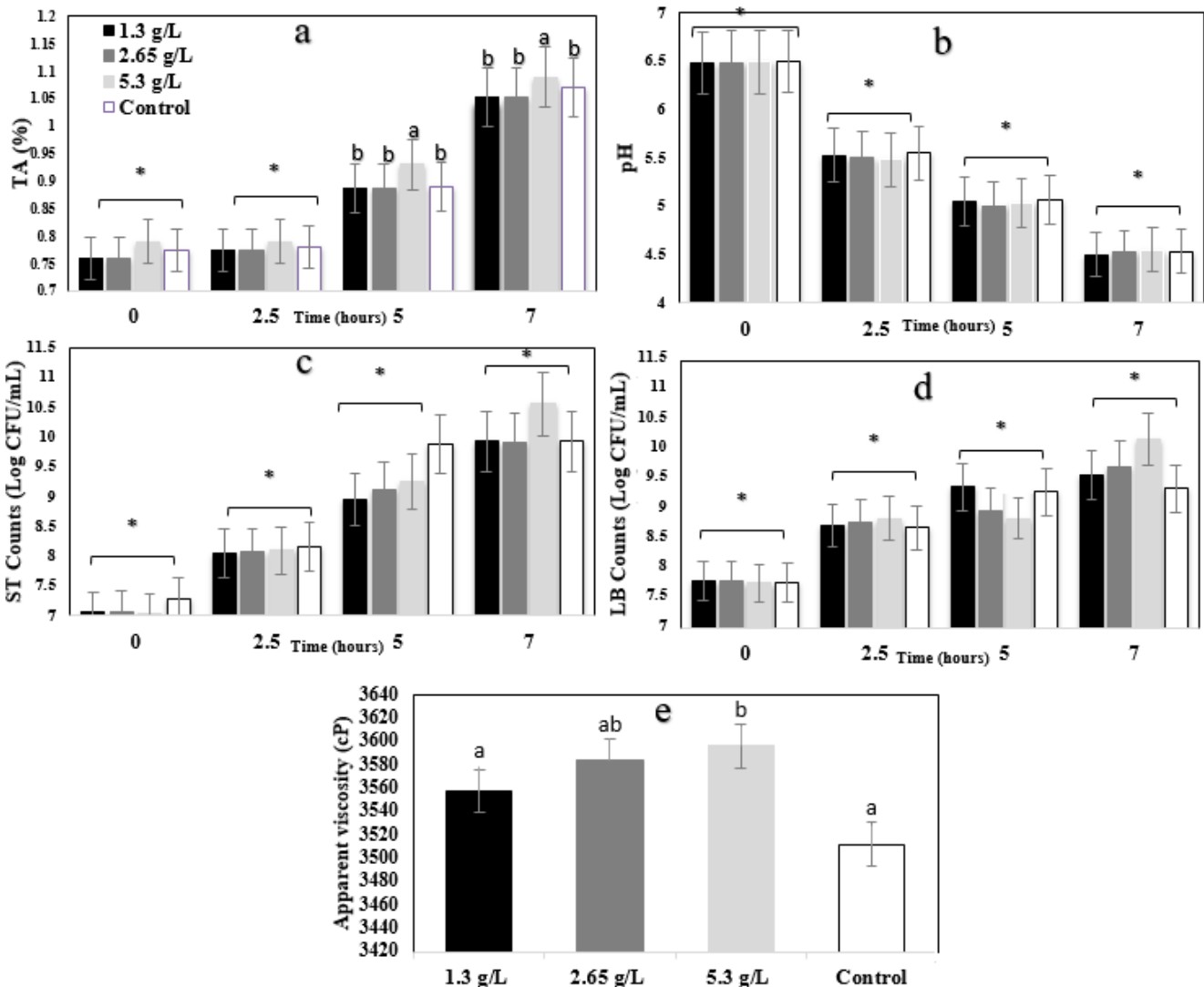

**Figure 1.** (**a**) Titratable acidity (TA) %, (**b**) pH, (**c**) *S. thermophilus* (ST) counts (Log CFU/mL), (**d**) *L. bulgaricus* (LB) counts (Log CFU/mL), and (**e**) apparent viscosity of control (0 g/L) and carao yogurt (CY) (1.3 g/L, 2.65 g/L and 5.3 g/L) during 0, 2.5, 5, 7 h of fermentation. * Indicates that treatments were not significantly ($p > 0.05$) different within control yogurt and CY samples. [a,b] Means with different letters within the treatments represent significant differences ($p < 0.05$) among control yogurt and (CY) samples.

Acidity and pH are the two most important parameters of fermented products, leading to estimated shelf-life. The bacterial starters decompose the lactose into lactic acid, resulting in acidity. Additionally, coagulation in the fermentation process of the yogurt is related to the decrease in pH and rise in total acidity. Similar results were observed in previous studies [44,45]. Although there is no significant difference ($p > 0.05$) between samples and control at 0 and 2.5 h for TA values, control and treated samples with carao powder demonstrated a significant difference in 7 h of incubation. A significant ($p < 0.05$) decrease in pH and an increase in acidity were shown over time. The pH was not significantly affected by adding the carao powder, but the total acidity of 5.3 g/L CY was higher than other samples.

Figure 1c,d illustrate *S. thermophilus* (ST) and *L. bulgaricus* (LB) counts of YS during 7 h of fermentation. For both bacteria, the time effect was significant ($p < 0.05$), whereas the concentration effect and the concentration * time interaction effect were not significant ($p > 0.05$). The ST and LB counts increased significantly ($p < 0.05$) during 0, 2.5, 5, and

7 h of fermentation. Carao powder did not significantly ($p > 0.05$) affected the ST and LB counts for 7 h. Although 5.3 g/L yogurt showed a slightly higher value (8.5 and 9 Log CFU/mL), there were no significant ($p > 0.05$) differences among the samples. The ST and LB counts increased gradually, indicating progressive fermentation. It is reported that using bioactive compounds and tropical fruits' pulps such as guava or soursop can maintain the bacterial viability of *Lactobacillus rhamnosus* and *Bifidobacterium animalis* after the first day of fermentation, which is due to the bioactive and bactericidal compounds [46]. During the fermentation, the viable counts of ST and LB in CY samples were above the 6.0 Log CFU/mL, indicating that carao does not harm the growth of yogurt starter culture bacteria. It is important that functional ingredients do not interfere with microbial growth in dairy fermented products.

The apparent viscosity of yogurt samples (1.3, 2.65, and 5.3 g/L) are shown in Figure 1e. Apparent viscosity significantly ($p < 0.05$) increased by adding carao powder at 5.3 g/L when compared to control yogurt. Galactomannan has been found in carao samples [20], and it has been shown to have a great ability to bind water [47]. It is possible that this polysaccharide could increase viscosity in the CY samples. Adding bioactive molecules to yogurt could also play a decisive role in the texture of fermented milk [48].

### 3.2. Total Polyphenols (TPC), Total Flavonoid Content (TFC), and Antioxidant Activity (AA) Evaluations

Table 1 shows the total polyphenols (TPC), total flavonoid content (TFC), and antioxidant activity (AA) values of the yogurt samples. There were no significant difference ($p < 0.05$) between different CY (1.3, 2.65, and 5.3 g/L) and control samples in AA. The fact that there was no significant antioxidant activity is notice. The content of total polyphenols and flavonoids in 5.3 g/L CY was significantly higher than in control and 1.3 g/L yogurt samples. Carao has a high flavonoid content [20,22]. One primary purpose of plant-based fortification into dairy cultured products is to increased antioxidant activity. Plant-based ingredients are a good source of polyphenols and have a higher phenolic content when compared to animal-based components.

**Table 1.** Effect of carao concentration on the total phenolic content (TPC), total flavonoid content (TFC), and antioxidant activity of freeze-dried yogurt.

| Level of Carao Extract | TPC (µg GAE/g) | TFC (µg Q/g) | Antioxidant Activity (%) |
|---|---|---|---|
| Control (0 g/L) | 13.54 ± 1.03 [a] | 4.32 ± 0.88 [b] | 2.5 ± 0.67 [a] |
| CY 1.3 g/L | 13.46 ± 1.01 [a] | 5.23 ± 0.47 [b] | 2.5 ± 0.69 [a] |
| CY 2.65 g/L | 14.03 ± 1.02 [ab] | 6.38 ± 0.42 [a] | 2.6 ± 0.72 [a] |
| CY 5.3 g/L | 14.41 ± 0.89 [b] | 7.07 ± 0.55 [a] | 2.7 ± 0.88 [a] |

[a,b] Means with different letters within the same column represent significant differences at $p < 0.05$ among control yogurt and (CY) samples.

### 3.3. Alpha-Glucosidase and Lipase Inhibition Evaluation

$\alpha$-Glucosidase and pancreatic lipase digest carbohydrates and lipids, respectively [49]. The activity of enzymatic inhibition was observed in how carao powder regulates alpha-glucosidase and lipase activity. Inhibition of $\alpha$-glucosidase and lipase activity is shown in Figure 2. The most effective CY for lipase inhibition activity (52.11%) was 2.65 g/L CY, which was 6.62% higher than control yogurt, and $\alpha$-glucosidase inhibition was more effective in 5.3 g/L CY (36.11), which was 10.53% higher than the control. Plants can inhibit $\alpha$-glucosidase and lipase activity to prevent diabetic complications such as hyperglycemia (diabetes) and hyperlipidemia (metabolic syndrome and heart disease) due to their antioxidant contents and bioactive compounds such as flavonoids [50,51]. Free phenolics of a peanut meal have been shown to inhibit $\alpha$-glucosidase and pancreatic lipase [48]. Other plants such as Dendrobium, genus of mostly epiphytic and lithophytic orchids in the family *Orchidaceae*, also have $\alpha$-glucosidase and lipase activity and can also be used for the treatment of diabetes [52]. Catechins and catechin gallates have been found in carao and

have been reported to inhibit activity against maltase. Polyphenols have been reported to have enzymatic inhibitory actions toward lipase and glucosidase activities [53].

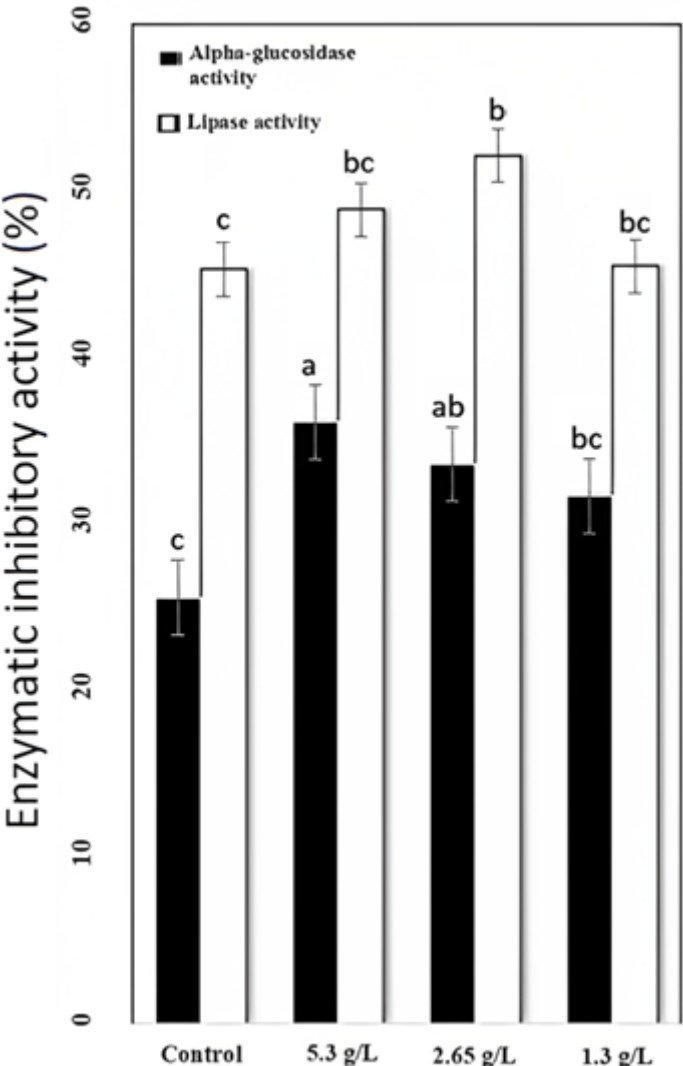

**Figure 2.** Effect of carao powder on the inhibition of α-glucosidase (black bar) and lipase (white bar) activity (%) on 0 g/L yogurt (control) and carao yogurt (CY) (1.3, 2.65, and 5.3 g/L). Data represent mean values for each sample ± standard deviation ($n$ = 3). [a–c] Means with different letters within the same enzymatic activity show differences among control yogurt and CY samples ($p < 0.05$).

### 3.4. Transepithelial Electrical Resistance and Paracellular Permeability

Transepithelial electrical resistance (TEER) was applied to measure cell monolayer integrity. The results are shown in Figure 3. All YS and control yogurt changed the function of Caco-2 cells treated with an inflammatory stimulus. The inflammatory stimulus mixture induced a 52.71% decrease in TEER value in the inflammation-induced sample. Decrease in TEER value in the sample with the inflammatory stimulus (interleukin-1β, tumor necrosis factor-α, interferon-gamma, LPS, and isoflavone genistein) (242.73 Ω·cm$^2$) happened due to disruptions to barrier integrity after 72 h (Figure 3a). Consequently, intestinal protection dysfunction was induced in Caco-2 cells by applying the inflammatory stimulus. Yogurt samples did not affect the TEER values in the first 24 h, but the TEER significantly ($p > 0.05$) increased after 48 h. The 5.3 g/L CY had a higher effect on TEER value up to 506.76 Ω·cm$^2$, compared to 1.3 and 2.65 g/L CY. Flavonoid and phenolic compounds of carao fruit could inhibit an induced barrier dysfunction after 24 h. There was no significant difference ($p < 0.05$) between TEER of control and 1.2, 2.65 5.3 g/L CY after 72 h.

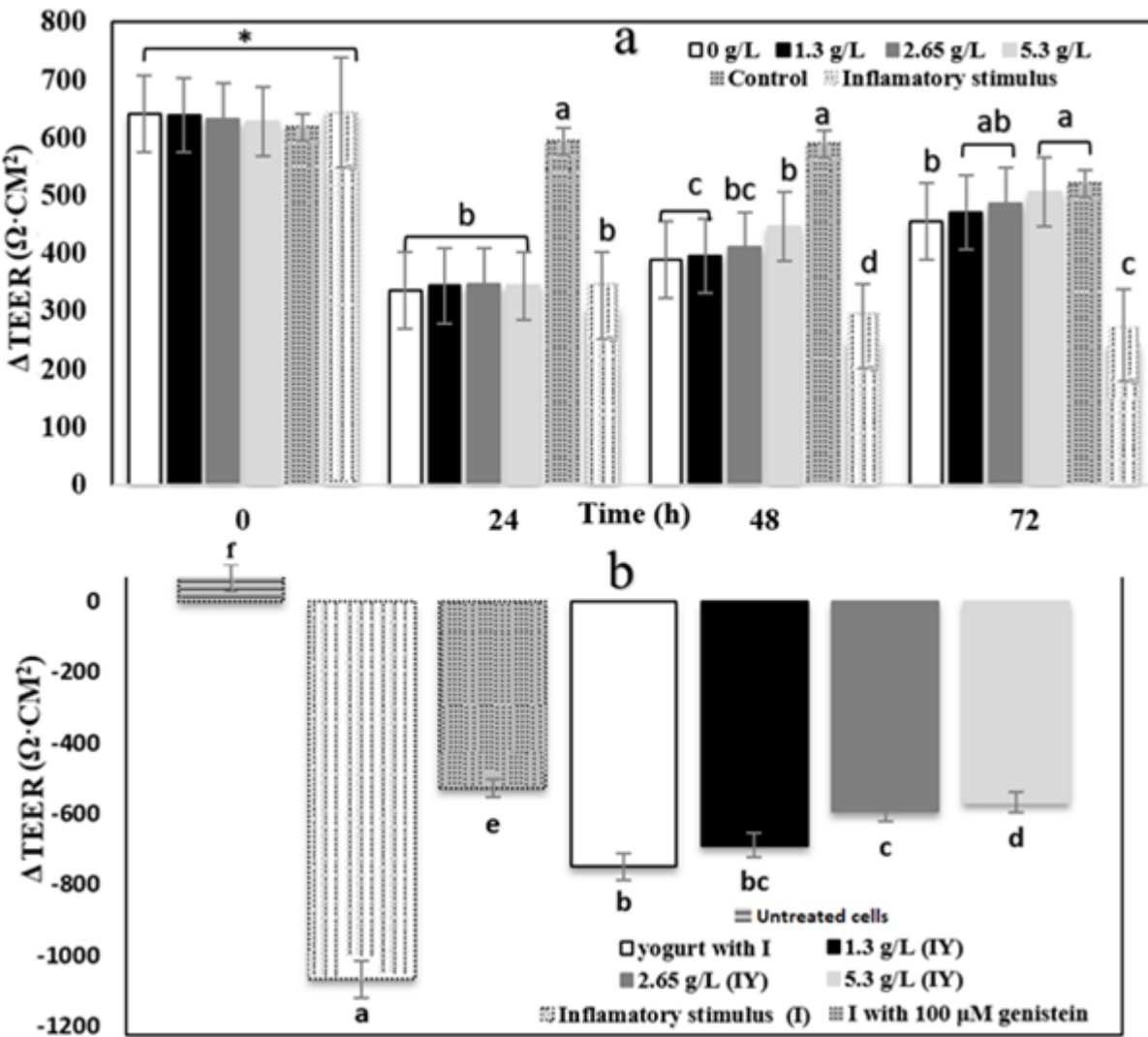

**Figure 3.** Carao yogurts (CYs) raise the transepithelial electrical resistance (TEER) of Caco-2 cell monolayers exposed to an inflammatory stimulus. (**a**) Caco-2 cells treated with vehicle control, inflammatory stimulus consisting of 25 ng. mL$^{-1}$ of IL-1β, 50 ng. mL$^{-1}$ of TNF-α, and 50 ng. mL$^{-1}$ of IFN-γ, or I and CYs from 0 to 72 h. (**b**) Caco-2 cells treated with CYs, control, inflammatory stimulus (I), I with 100 μM genistein for 48 h. * indicates that the treatments were not significantly (*p* > 0.05) different among control yogurt, CY samples, inflammatory stimulus, and control cells for 0, 24, 48, and 72 h. $^{a-f}$ different letters indicates significant differences in TEER values among treatments.

Δ TEER (differential) of yogurt with 100 μM genistein was -526.17 Ω·cm$^2$ (Figure 3b), indicating that it significantly (*p* > 0.05) decreased TEER values when compared to IS samples having less inflammation. According to the results, control and CY could decrease the inflammation in Caco-2 cells while the most effective treatment among the others was 5.3 g/L CY (−567.36 Δ TEER Ω·cm$^2$). An increase in the permeability of fluorescein isothiocyanate dextran 4000 (FD, Figure 4a, and LY, Figure 4b; ug/mL) and a decrease in TEER (Figure 2) by cells treated with only interleukin-1β, tumor necrosis factor-α, interferon-gamma, LPS, and isoflavone genistein illustrates that the inflammatory stimulus damaged the integrity of the Caco-2 cell monolayer [54]. Tight junction proteins, occludin, and (ZO)-1 decrease resulted from a disruption of the intestinal barrier, which was induced by inflammatory stimulus [54]. It is likely that a change in the mRNA expression of proteins caused the barrier function improvement by adding yogurt and carao powder [36]. In relation to this, rich polyphenolic ingredients are desirable for the fortification of cultured

dairy products to improve the products' functionality. Polyphenols are known to moderate the overexpression of adhesion molecules (VCAM-1 and ICAM-1) by inhibiting the NF-κB pathway in endothelial cells [55].

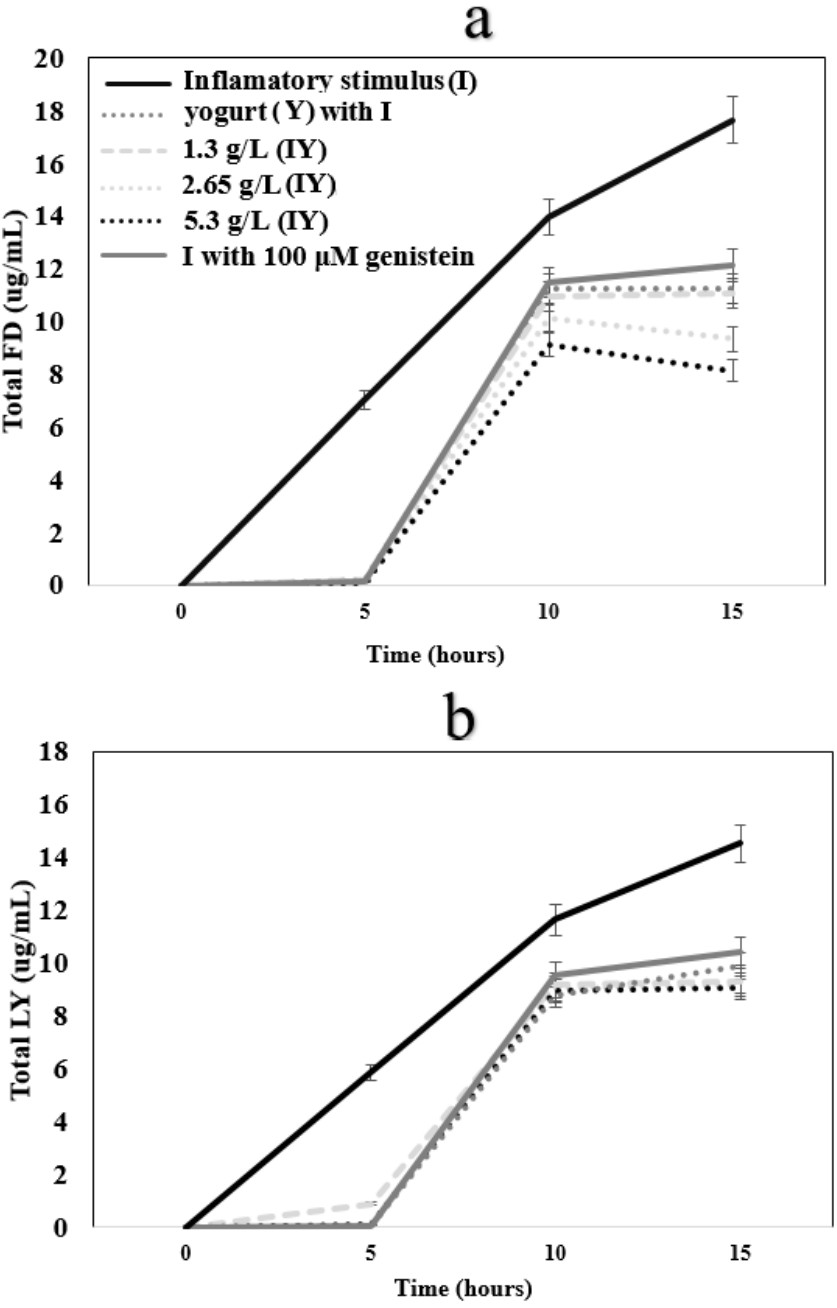

**Figure 4.** The flux of (**a**) fluorescein isothiocyanate-dextran (FD) and (**b**) Lucifer yellow (LY) in differentiated Caco-2 cells exposed to control (healthy cells), inflammatory stimulus (I), or inflammatory stimulus and carao yogurt samples (CYs) for 15 h.

*3.5. Transmission Electron Microscopy*

Transmission electron microscopy (TEM) images are shown in Figure 5, illustrating the effect of carao powder in Caco-2 cell monolayer tight junctions' integrity. The presence of electron-dense material (black streaks or electron-dense areas in Figure 5F) in the space between cells near the brush border reflects the tight junctions [37]. In cells treated with only the inflammatory stimulus (Figure 5A), the tight junctions, as indicated by the area of black streaks, were less when compared to cells treated with control cells (healthy cells)

(Figure 5F), control yogurt (Figure 5B), and CY (Figure 5C–E). CY (Figure 5C–E) contained more tight junctions (electron-dense material or black streaks) when compared to cells treated with control yogurt (Figure 5B). In TEM images, the tight junctions in the 5.3 g/L of CY samples (Figure 5E) were larger than the other YS (Figure 5B–E). The TEM images are related to the TEER and paracellular permeability measurements, where 5.3 g/L of CY reduces inflammation when compared to other the treatments by showing the integrity of the tight junctions. The increased tight junctions' intact structure from CY samples could possibly be due to the protein content and phytochemicals in the carao [56].

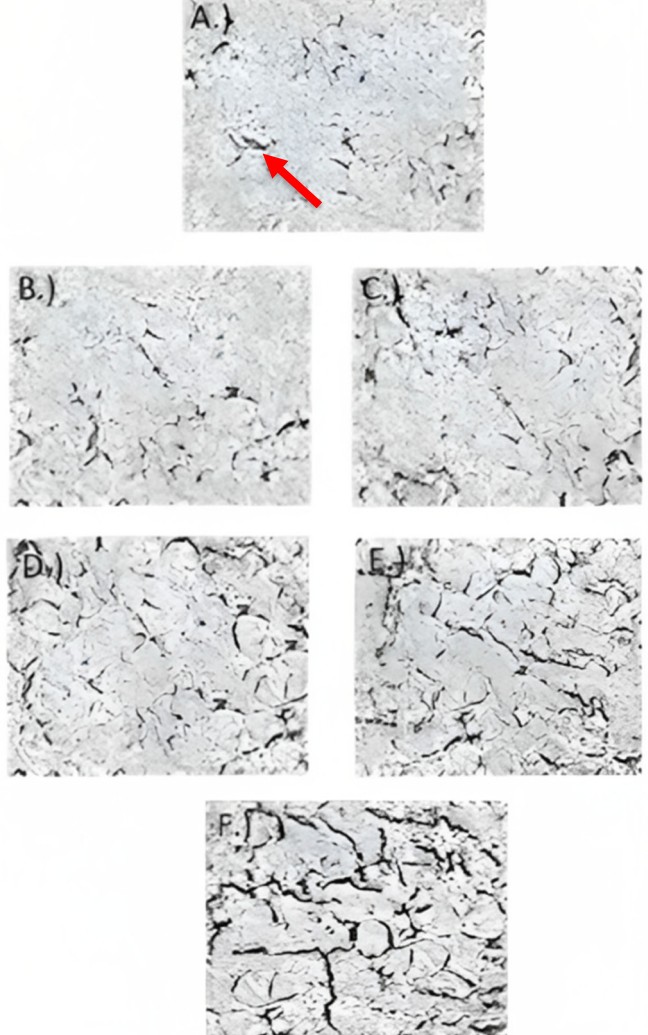

**Figure 5.** Transmission electron microscopy images of Caco-2 cells treated with (**A**) inflammatory stimulus (IS) consisting of interleukin-1β, tumor necrosis factor-α, interferon-gamma, LPS, and isoflavone genistein, (**B**) control yogurt (0 g/L with inflammatory stimulus (IS)), (**C**) 1.3 g/L CY with IS, (**D**) 2.6 g/L CY with IS, (**E**) 5.3 g/L CY with IS, and (**F**) control cells (untreated cells) during 72 h. Pictures were taken under approximately 90 nm$^2$. Red arrow indicates black streaks or electron-dense area.

*3.6. Immunofluorescence Microscopy*

For localizing ZO-1 (Figure 6), occludin-1 (Figure 7), and claudin-1 (Figure 8) in fully differentiated Caco-2 cells, different treatments and control were observed with immunofluorescence microscopy. ZO-1 is dispersed vertically throughout the cytoplasm. The presence of net-green patterns between cells for ZO-1 were detected on the immunofluorescence microscopy photos which confirmed that the cells were not merged, showing fully func-

tional tight junctions. The presence of net-green patterns between cells (red arrows in Figure 6F) in the space between cells near the brush border reflects the ZO-1 tight junction's intact structure, similar to previous studies [34,37]. On the other hand, in cells with the inflammatory stimulus, the ZO-1 green patterns appear less preserved (Figure 6A), and a significantly ($p > 0.05$) lower florescence intensity (Figure 9) was observed when compared to cells-treated YS (Figure 6B–E) and control cells (healthy cells) (Figure 6F). Similarly, occludin-1 (Figure 7A–F), which is found at the lateral apical end of the cytoplasm, showed a similar trend to ZO-1. The claudin-1 and occludin-1 relative intensity/cell were improved by 5.3 g/L carao yogurt having a higher florescence intensity when compared to control cells (Figure 9b,c).

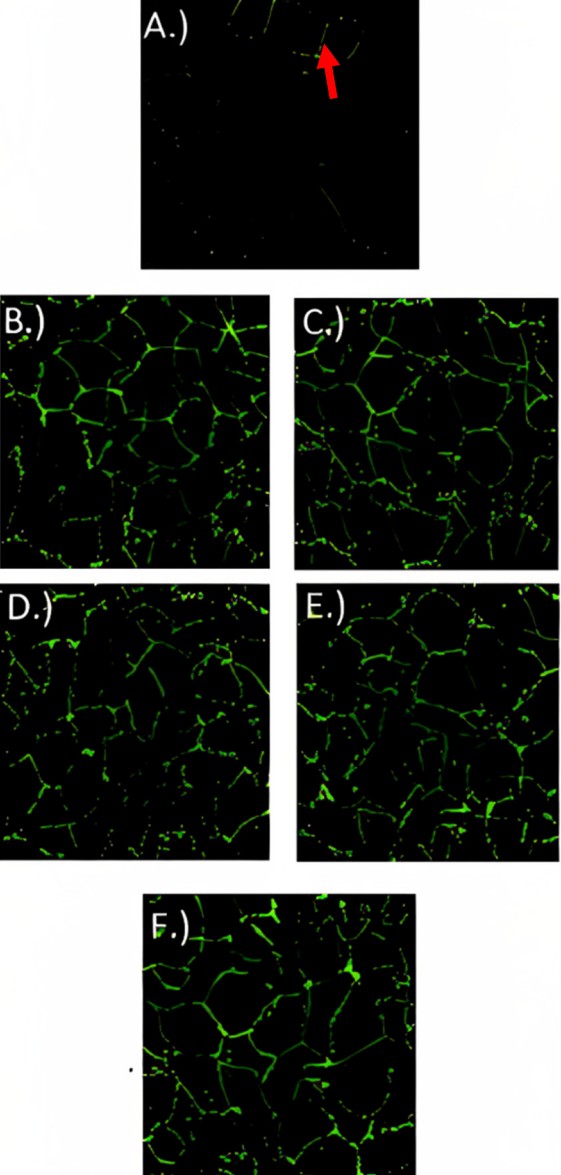

**Figure 6.** ZO-1 immunofluorescence microscopy pictures of Caco-2 cells treated with (**A**) inflammatory stimulus (IS) consisting of interleukin-1β, tumor necrosis factor-α, interferon-gamma, LPS, and isoflavone genistein; (**B**) control yogurt (0 g/L with IS; (**C**) 1.3 g/L CY with IS; (**D**) 2.6 g/L CY with IS; (**E**) 5.3 g/L CY with IS; and (**F**) control cells (untreated cells) during 72 h. Pictures are taken under approximately 75 μm². Red arrow indicates net-green patterns between cells (fluorescence intensity).

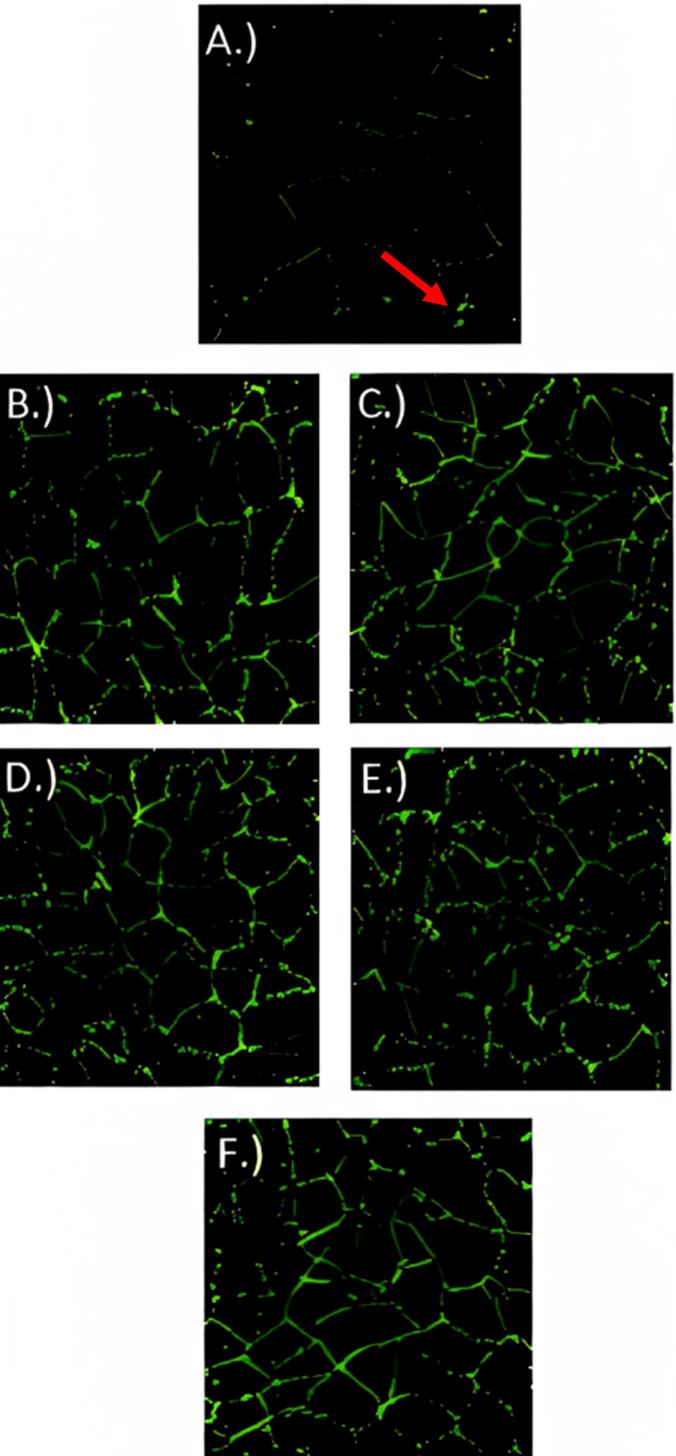

**Figure 7.** Occludin-1 immunofluorescence microscopy pictures of Caco-2 cells treated with (**A**) inflammatory stimulus (IS) consisting of interleukin-1β, tumor necrosis factor-α, interferon-gamma, LPS, and isoflavone genistein; (**B**) control yogurt (0 g/L with inflammatory stimulus (IS)); (**C**) 1.3 g/L CY with IS; (**D**) 2.6 g/L CY with IS; (**E**) 5.3 g/L CY with IS; and (**F**) control cells (untreated cells) during 72 h. Pictures were taken under approximately 75 μm². Red arrow indicates net-green patterns between cells (fluorescence intensity).

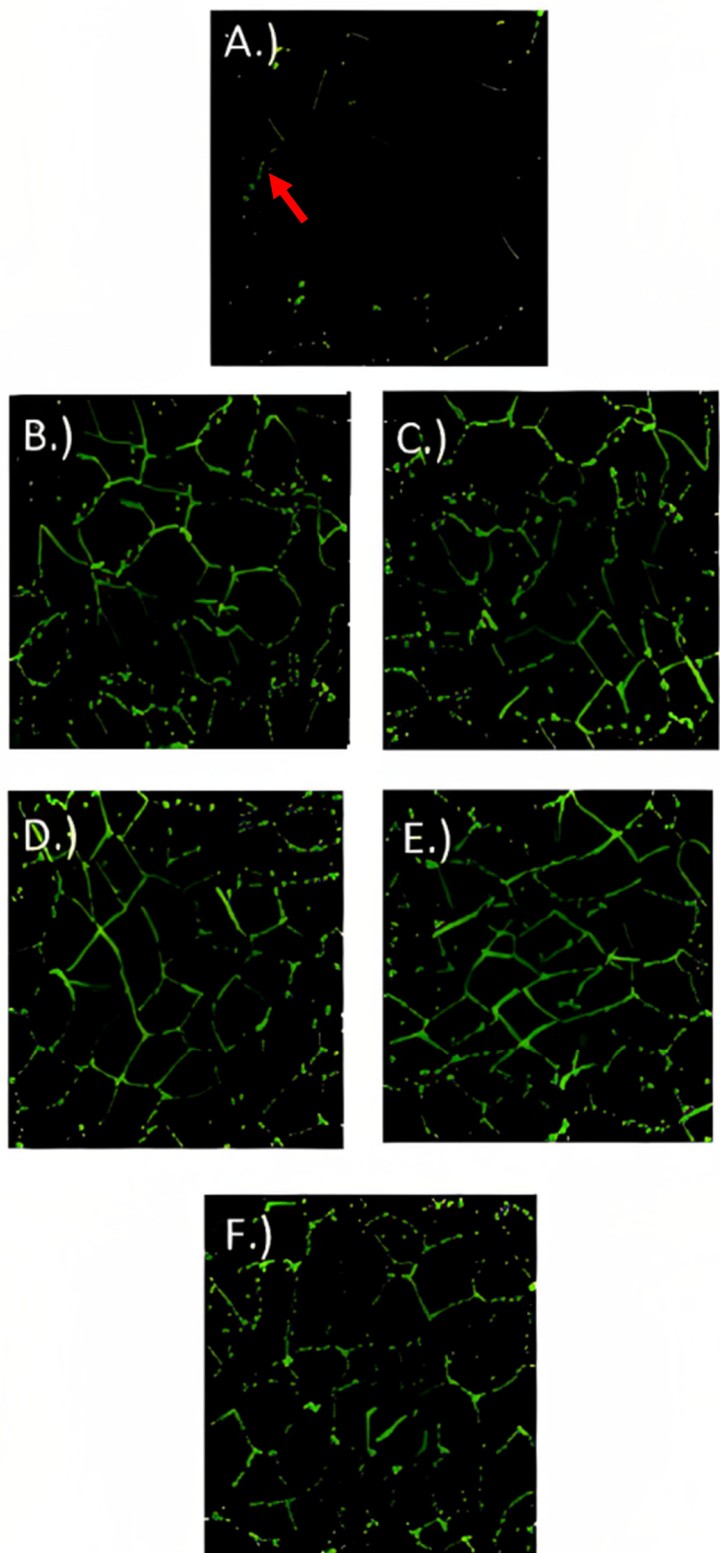

**Figure 8.** Claudin-1 immunofluorescence microscopy pictures of Caco-2 cells treated with (**A**) inflammatory stimulus (IS) consisting of interleukin-1β, tumor necrosis factor-α, interferon-gamma, LPS, and isoflavone genistein; (**B**) control yogurt (0 g/L with inflammatory stimulus (IS)); (**C**) 1.3 g/L (CY) with IS; (**D**) 2.6 g/L CY with IS; (**E**) 5.3 g/L CY with IS; and (**F**) control cells (untreated cells) during 72 h. Image size is approximately 75 μm². Red arrow indicates net-green patterns between cells (fluorescence intensity).

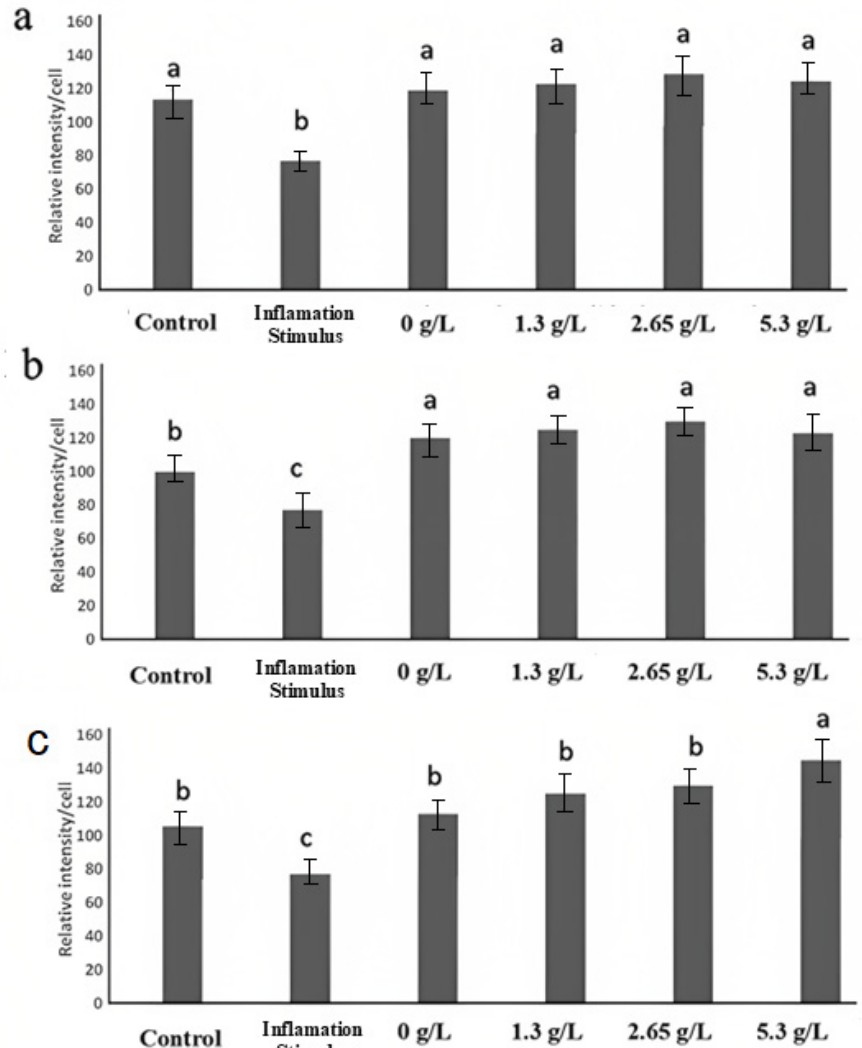

**Figure 9.** (**a**) ZO-1, (**b**) claudin-1, and (**c**) occludin relative intensity/cell of control (healthy and untreated cells with no inflammatory stimulus), inflammatory stimulus (IS) (cells treated with only IL-1β, TNF-α, IFN-γ, LPS, and isoflavone genistein), regular yogurt (0 g/L) with IS, and CY (1.3, 2.65, 5.3 g/L) with IS. [a–c] Different letters denote significant differences between groups at $p < 0.05$ among control yogurt and CY samples.

On the other hand, the claudin-1 (Figure 8A–F) was distributed tightly through the cell and it was found to have a higher florescence intensity in 5.3 g/L CY samples (Figure 8E) (Figure 9c) when compared to the control yogurt (Figure 8B) and control cells (healthy cells) (Figure 8F). Similar to occludin-1 and ZO-1, cells treated with the inflammatory stimulus decreased the staining of claudin-1 tight junction proteins [34], where fewer net-green patterns were observed when compared to cells-treated YS (Figure 8B–E). Figures 6–8 confirm how carao powder impacts the functional tight junctions with steady mesh-like patterns among adjoining cells, demonstrating its localization and integrity. Different studies have tested the effectiveness of various probiotic bacteria in improving tight junctions, including *Lactobacillus rhamnosus* GG, *Lactobacillus acidophilus*, *Lactobacillus Plantarum*, *Bifidobacterium infantis*, *Bifidobacterium animalis lactis* BB-12, and *Escherichia coli Nissle* 1917 [57]. The immunofluorescence microscopy observations confirmed the observations of transmission electron microscopy (Figure 5), where cells treated with YS improved the tight junctions. There are very few studies regarding the effects of yogurt on gut inflammation. The consumption of whole and skimmed yogurt has been associated with an anti-inflammatory effect in subjects with metabolism disorders, including obesity and overweight [58]. The

study by Meng et al. (2017) [59] shows the disparity in its effects on markers of inflammation since the consumption of yogurt improves some markers, such as TLR-2 in monocytic cells (CD4 + HLA − DR+). Nevertheless, it does not modify the levels of TNF-$\alpha$ or IL-6 in peripheral blood mononuclear cells (PBMC) stimulated in vitro. In contrast, the study by Meyer et al. (2007) [60] suggests a possible role for yogurt in the stimulation of the immune system against infections, since an increase in TNF-$\alpha$ and IL-1$\beta$ levels in PBMC stimulated in vitro with LPS or phytohemagglutinin (PHA), while the study by Pei et al. (2017) [61] shows a reduction in the levels TNF-$\alpha$ plasma levels and an enhancement of endotoxemia markers (LPS binding protein (LBP)/sCD14) compared to the control group, without affecting the levels of IL-6 and protein C reagent (PCR).

### 3.7. Gene Expression Analysis of Tight Junction Proteins

The real-time quantitative polymerase chain reaction was applied to examine if yogurt fortified with carao improved the gene expression of tight junction proteins occludin, ZO-1, and claudin-1 in Caco-2 cells due to their significance in modulating intestinal barrier functions. The relative expression of occludin, ZO-1, and claudin-1 is shown in Figure 10. When Caco-2 cells were treated with only inflammatory stimulus (IL-1$\beta$), (TNF-$\alpha$ (IFN-$\gamma$)), (LPS), and isoflavone genistein), the relative expression of occludin and ZO-1 decreased when compared to control (cells with no inflammatory stimulus). The levels of the relative expression of ZO-1 for YS did not differ when compared to the control. On the other hand, the levels of relative expression of occludin and claudin-1 increased with yogurt fortified with 2.65 g/L and 5.3 g/L CY when compared to control samples. The most notable increase in claudin-1 gene relative expression was observed when cells were treated with 5.3 g/L CY. Previous studies have also found that occludin and ZO-1 gene relative expressions do not change noticeably when they are damaged due to inflammation [62]. The increase in the relative expression of occludin and claudin-1 in yogurt fortified with 2.65 g/L and 5.3 g/L carao and the inflammatory stimulus were related to the observations of transmission electron and immunofluorescence microscopy (Figures 5–8). Furthermore, the paracellular permeability and transepithelial electrical resistance show that yogurt fortified with carao decreases intestinal barrier dysfunction by improving the tight junction proteins.

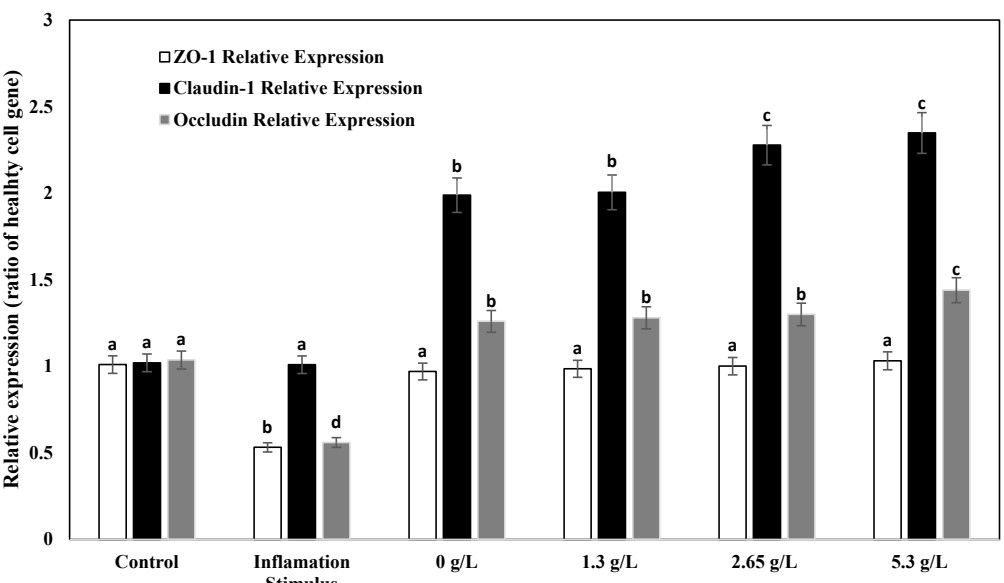

**Figure 10.** ZO-1 (white bar), claudin-1 (black bar), and occludin (gray bar) relative expression of control (healthy and untreated cells with no inflammatory stimulus), inflammatory stimulus (IS) (cells treated with only IL-1$\beta$, TNF-$\alpha$, IFN-$\gamma$, LPS, and isoflavone genistein), regular yogurt (0 g/L) with IS, and CY (1.3, 2.65, 5.3 g/L) with IS. [a–d] Different letters denote significant differences among groups (ZO-1, claudin-1, and occludin) at $p < 0.05$ among control yogurt and CY samples.

### 3.8. Fasting Blood Glucose and Oral Glucose Tolerance

Normal plasma glucose concentration is represented in Table 2. Fasting blood glucose varied from $4.02 \pm 0.45$ mmol/L at day zero to $6.33 \pm 0.78$ mmol/L at 42 days in healthy rats. The diabetic group had the highest blood sugar ($15.13 \pm 1.35$ mmol/L) after 42 days. Blood sugar increase results in alloxan's toxic effect, which reduces glucose-induced insulin secretion via glucokinase activity inhibition, resulting in insulin-dependent diabetes and beta cell necrosis [63]. The metformin group, which decreases the intestinal absorption of glucose and hepatic glucose production, had the highest effect on the decrease in blood sugar and showed a significant difference ($p < 0.05$) from the control group and CY samples. Among the control yogurt and CY samples, the most effective treatment was 5.3 g/L yogurt, showing the highest reduction in blood sugar after 42 days of feeding. Al-Salami et al. (2008) [64] reported that probiotic products had no effects on the blood sugar of healthy rats while having a significant effect on diabetic rats' blood sugar due to an increase in gliclazide bioavailability, which could improve glucokinase activity. There are other fruit extracts such as monk fruit that, when added to yogurt, are used as natural sweeteners and have positive effects on glucose regulation in type 2 DM [65]. The possible mechanisms of action could be associated with the β-cell function integrity, enhanced gut microbiota, and decreased insulin resistance. Other works such as the inclusion of honey in yogurt improved type 2 diabetes [66].

**Table 2.** Effects on fasting blood glucose (mmol/L) after feeding rats yogurt fortified with carao.

| Group | Fasting Blood Glucose (mmol/L) | | | |
|---|---|---|---|---|
| Time After Feeding (Days) | 0 | 14 | 28 | 42 |
| Control (healthy rats) | $4.0 \pm 0.54$ [a] | $4.74 \pm 0.67$ [a] | $4.95 \pm 0.84$ [a] | $6.33 \pm 0.78$ [a] |
| Diabetic rats | $4.2 \pm 0.63$ [a] | $4.48 \pm 0.41$ [a] | $4.83 \pm 0.67$ [a] | $15.13 \pm 1.35$ [f] |
| 0 g/L | $3.9 \pm 0.36$ [a] | $4.59 \pm 0.59$ [a] | $4.90 \pm 0.29$ [a] | $13.03 \pm 1.37$ [e] |
| 1.3 g/L | $4.2 \pm 0.50$ [a] | $4.45 \pm 0.37$ [a] | $4.59 \pm 0.73$ [a] | $11.07 \pm 1.29$ [de] |
| 2.65 g/L | $4.1 \pm 0.42$ [a] | $4.68 \pm 0.34$ [a] | $4.83 \pm 0.80$ [a] | $10.05 \pm 1.83$ [cd] |
| 5.3 g/L | $4.2 \pm 0.77$ [a] | $4.77 \pm 0.51$ [a] | $4.69 \pm 0.42$ [a] | $9.16 \pm 1.05$ [c] |
| Metformin | $4.1 \pm 0.49$ [a] | $4.42 \pm 0.27$ [a] | $4.64 \pm 0.55$ [a] | $7.56 \pm 0.96$ [b] |

[a–f] Different lowercase letters within a column denote significant differences between groups at $p < 0.05$. Control = rats fed a normal diet and water; diabetes control group with 30 mg/mL sucrose; control yogurt group containing 0 mg/mL of carao; carao yogurt (CY) group with 1.3 g/L carao; 2.65 mg/mL CY group, 5.3 mg/mL CY group; metformin group, 300 mg/kg metformin. On the first day of week 6, the diabetes group, control yogurt group, 0 mg/mL of carao; 1.3 g/L CY; 2.65 mg/mL CY 5.3 mg/mL CY. All diabetic-induced rats received streptozotocin freshly dissolved in 50 mmol/L citrate buffer (pH 4.5) at a dose of 100 mg/kg of body weight.

An analysis of AUC (area under the curve for glucose tolerance over time) indicates significant differences ($p < 0.05$) between treatments and control (healthy rats) (Figure 11a). Among yogurt samples, 2.65 g/L and 5.3 g/L had the higher values, leading to conclude that the glycemic response of the yogurt sample groups significantly improved with carao fortification [67].

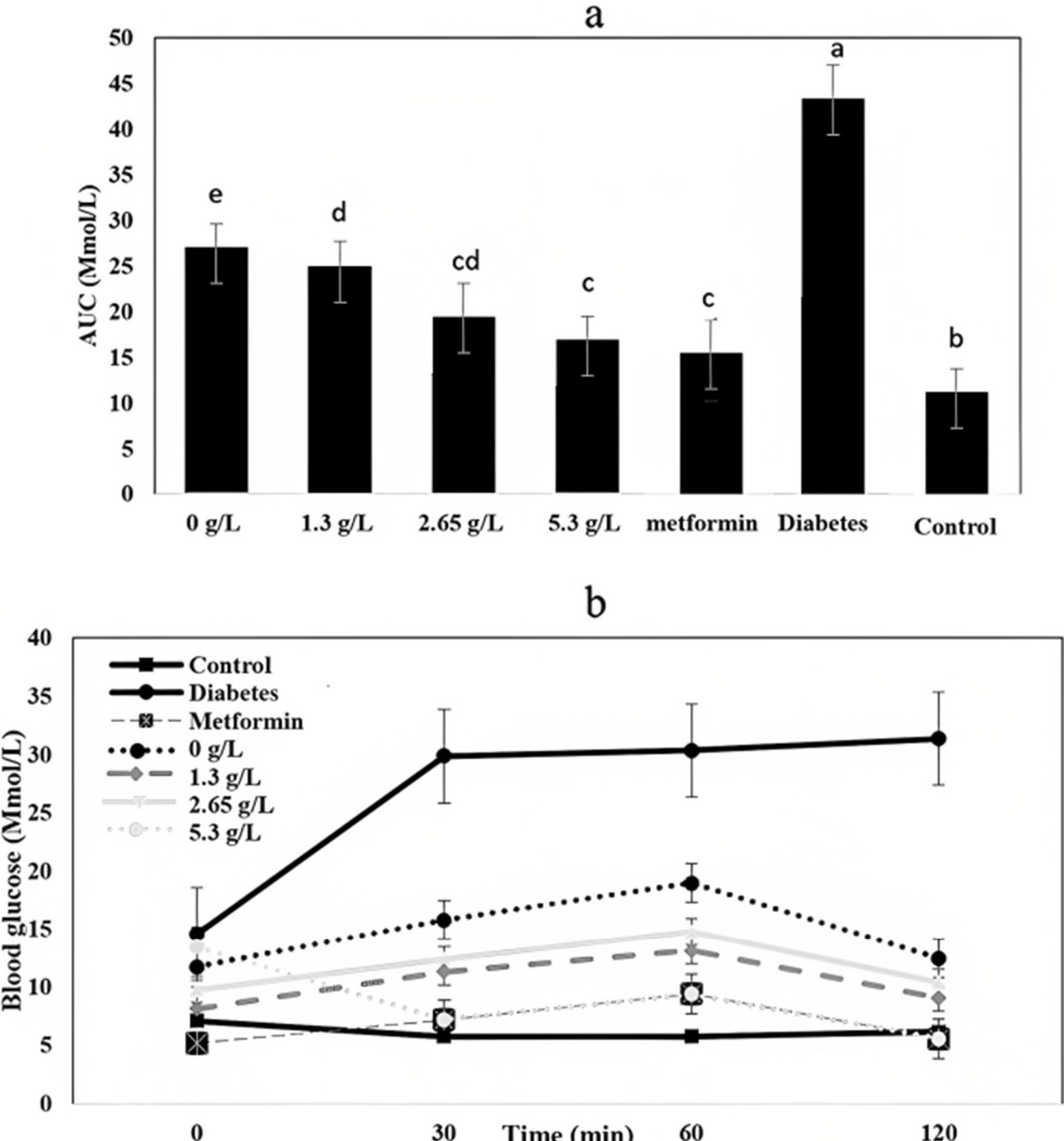

**Figure 11.** (**a**) AUC (area under the curve for glucose tolerance over time) (Mmol/L) of healthy rats (control) yogurt rats (0, 1.3, 2.65, and 5.3 g/L), diabetic rats (diabetes), and metformin rats. [a–e] Means with different letters show differences among samples (*p* < 0.05). (**b**) Effect of carao yogurt (CYs) (0, 1.3, 2.65, 5.3 g/L) and controls on oral glucose tolerance test (mmol/L) from 0 min to 120 min.

The oral glucose tolerance test values are shown in Figure 11b. Glucose concentration for carao yogurt at 5.3 g/L dramatically decrease at 30 min and then slowly decreased to the initial level, whereas the diabetic group increased up to 30 Mmol/L at 120 min (Figure 11b). Yogurt has shown to normalize the glucose levels in high-fat diet-induced metabolic syndrome and oxidative stress of the obese rats to prevent glucose fluctuation [68]. In some studies, carao has shown great antidiabetic potential in animal models and improves the attributes of probiotic bacteria such as *L. acidophilus* [69,70]. For this reason, it is

recommended for fortification purposes in cultured dairy products when trying to develop a product for consumers with leaky gut and diabetic complications.

## 4. Conclusions

Intestinal barrier dysfunction is associated with inflammatory bowel disease, obesity, and leaky gut syndrome. Overall, results indicated that yogurt fortified with carao had a positive effect on pH, apparent viscosity, total flavonoid content, TEER ($\Omega \cdot cm^2$), LY and FD (ug/mL), $\alpha$-glucosidase activity, and lipase inhibition activity, fasting blood sugar (Mmol/L), and AUC (Mmol/L). Functional effects of the formulation of yogurt supplemented with carao were also evaluated in terms of the antioxidant activity, the $\alpha$-glycosidase activity, and lipase inhibitory properties. In addition, the hypoglycemic potential was in vivo validated in a rat model. Compared to the control yogurt, Caco-2 TEER (transepithelial electrical resistance evaluation) by yogurt with 5.3 g/L of carao was significantly lower ($p < 0.05$) after 48 h. Yogurt with 5.3 g/L of carao had a considerably lower permeability ($p < 0.05$) than control yogurt in FD and LY flux. The main reason for these effects might be due to the flavonoid and bioactive content of the carao pulp, which improved the characteristics of yogurt compared to the control yogurt sample. To sum up, yogurt fortified with carao improves the integrity of Caco-2 cells by strengthening tight junctions and has a predominant hypoglycemic effect in rats. This study suggests fortifying yogurt with carao when improving the functionality of yogurt in terms of intestinal barrier functions and antidiabetic potential.

**Author Contributions:** Conceptualization, R.S.A. and R.P. methodology, R.S.A. and S.K.P. software, R.S.A., R.P. and S.K.P. formal analysis, R.S.A., J.M., D.M.-V., V.M.-F. and I.M.-F. investigation, R.S.A., R.P., J.M., D.M.-V and, S.K.P. resources, R.S.A., J.M., I.M.-F. and K.A. data curation, I.M.-F., R.S.A. and V.M.-F.; writing—original draft preparation, R.S.A., R.P., S.K.P., V.M.-F., D.M.-V. and I.M.-F. writing— review and editing, R.S.A., R.P., S.K.P.,D.M.-V., V.M.-F., K.A. and I.M.-F. project administration, R.S.A. and K.A. funding acquisition, R.S.A., K.A., R.S.A., I.M.-F. and J.M. All authors have read and agreed to the published version of the manuscript.

**Funding:** This research was funded by the Junta de Extremadura (ref. GR21121–AGA008), the European Regional Development Fund (FEDER), University National of Agriculture (Honduras) (Ref. C-DSIP-008-2023-UNAG), and USDA Hatch funds LAB94511.

**Institutional Review Board Statement:** The established animal experiment guidelines by the Declaration of Helsinki were used as a reference for the animal models, which is approved by the Ethics Committee in Food Research at the Honduran Association of Medicine and Nutrition (ASOHMENU) with form number AS-ASHOMENU-0013-2022.

**Informed Consent Statement:** Not applicable.

**Data Availability Statement:** The authors confirm that the data supporting the findings of this studyare available within the article and the raw data that support the findings are available from thecorresponding author, upon reasonable request. (PDF) Attributes of Lactobacillus acidophilus as Effected by Carao (Cassia grandis) Pulp Powder. Available from: https://www.researchgate.net/publication/370221110_Attributes_of_Lactobacillus_acidophilus_as_Effected_by_Carao_Cassia_grandis_Pulp_Powder (accessed 7 June 2023).

**Acknowledgments:** The authors thank the Research Support Service of the University of Extremadura. Finally, we also appreciate the support of the University National of Agriculture (Honduras) and the Louisiana State University Agricultural Center.

**Conflicts of Interest:** The authors declare no conflict of interest.

**Abbreviation**

| | |
|---|---|
| CY | Carao yogurt |
| DM | Diabetes mellitus |
| TA | Titratable acidity |
| LB | Lactobacillus bulgaricus |
| ST | Streptococcus thermophilus |
| DPPH | 2,2-Diphenyl-2-picrylhydrazyl |
| DMEM | Dulbecco's modified Eagle medium |
| TPC | Total phenolic content |
| TFC | Total flavonoid content |
| IL-1β | Interleukin-1β |
| TNF-α | Tumor necrosis factor-α |
| IFN-γ | Interferon-gamma |
| LPS | Lipopolysaccharide |
| TEER | Transepithelial electrical resistance evaluation |
| HBSS | Hank's balanced salt solution |
| FD | (FITC)-dextran |
| LY | Lucifer yellow |
| YS | Yogurt samples |
| NPG | p-Nitrophenyl-α-glucopyranoside |
| pNPP | p-Nitrophenol palmitate |
| AUC | The area under the curve |

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
