# Peer review of "Effects of Yogurt with Carao (Cassia grandis) on Intestinal Barrier Dysfunction, α-glycosidase Activity, Lipase Activity, Hypoglycemic Effect, and Antioxidant Activity"

_fermentation, doi:10.3390/fermentation9060566_

Round 1

Reviewer 1 Report

The work presented by Aleman and co-workers is about the potential of an aqueous Carao pulp extract as an active ingredient in yoghurts. This is an interesting study, in which several methodologies were applied.

In general, the most intriguing aspect of this manuscript is the lack of context, which is extremely serious in the M&M and Results sections. As general advice, I encourage the authors to consult a couple of papers available in the literature.

Abstract

line 21 - fix "in vitro" to the italic form, please

Keywords

Why do the authors consider "antiglycant" as a keyword? Which evidence is presented in this way?

Introduction

Lines 49-50: "complicated"? Do the authors want to mean multifactor??? Please redact this sentence.

Lines 56-60: Context is missing. Please redact the paragraph properly.

The aim of the work should also be presented in this section...

Material and methods

Most of the subsections belonging to this one lack references. Do all these methodologies applied for the first time or they are inspired by (several) previous ones? Sections 2.2, 2.4 (antioxidant and TFC), 2.8, 2.12, more precisely. In addition, almost none of them presents the number of replicates performed... The mention of the use of positive/negative controls is missing too.

Line 105: Please, present the full binomial form (at least the first time you present it).

Line 118: When the authors refer to "extract", should we understand "filtered" instead? if so, fix it, please.

Section 2.13: I'm encouraging the authors to take a look over some other papers available in the literature to redact this section properly. What is "Proc mixed"? the software used to perform the statistical analysis? If so, please explain it properly.

Results

Starting the presentation of the section with an acronym is just insane. Please, make a nice context mentioning why the methodologies chosen are important for the study... Saying that something was higher or lower is just describing what we can see in tables and figures. Science is about the meaning of things, the whys...

Q1: What were the criteria to define the ranging concentrations of Carao powder in yoghurts? 

Lines 336-339: Please, redact this part with some further context.

Line 347: A sentence with 5 words and 2 references?! Please, make an effort to improve this discussion.

Q2: Why the authors do not characterize the Carao extract by HPLC?

Author Response

Reviewer #1 Comments and Suggestions for Authors

The work presented by Aleman and co-workers is about the potential of an aqueous Carao pulp extract as an active ingredient in yoghurts. This is an interesting study, in which several methodologies were applied.

In general, the most intriguing aspect of this manuscript is the lack of context, which is extremely serious in the M&M and Results sections. As general advice, I encourage the authors to consult a couple of papers available in the literature.

Abstract

line 21 - fix "in vitro" to the italic form, please

Response: in vitro is in italic form.

Keywords

Why do the authors consider "antiglycant" as a keyword? Which evidence is presented in this way?

Response: Antiglycant was delete from keyword.

Introduction

Lines 49-50: "complicated"? Do the authors want to mean multifactor??? Please redact this sentence.

Response: This sentence was rephrased

Lines 56-60: Context is missing. Please redact the paragraph properly.

Response: context was added to the paragraph (Line 56-60). “The fortification of yogurt with functional ingredients can improve the functionality of yogurt.”

The aim of the work should also be presented in this section...

Response: As a result, the purpose of this study was to examine the effects of yogurt with carao on intestinal barrier dysfunctions in caco2 cells and hypoglycemic effect in animal model.

Material and methods

Most of the subsections belonging to this one lack references. Do all these methodologies applied for the first time or they are inspired by (several) previous ones? Sections 2.2, 2.4 (antioxidant and TFC), 2.8, 2.12, more precisely. In addition, almost none of them presents the number of replicates performed... The mention of the use of positive/negative controls is missing too.

Response: References were included in the methods. The use of positive/negative controls was mention and the number of replicates in the experiments was stated.

Line 105: Please, present the full binomial form (at least the first time you present it).

Response: Line 105 was restated as “Yogurts were serially diluted in sterilized peptone (0.1% wt/v) from 10-1 to 10-6 and pour plates in duplicate recording were incubated [27]…”

Line 118: When the authors refer to "extract", should we understand "filtered" instead? if so, fix it, please.

Response: The word extracted was changed to filtered.

Section 2.13: I'm encouraging the authors to take a look over some other papers available in the literature to redact this section properly.

Response: The statistical analysis was performed as “Aleman, R. S., Cedillos, R., Page, R., Olson, D., & Aryana, K. (2023). Physico-chemical, microbiological, and sensory characteristics of yogurt as affected by ingredients that help treat leaky gut. Journal of Dairy Science”.

What is "Proc mixed"?

Response: In our case study, mixed proc model random (Replications in yogurt batches) and mixed effect (treatments; carao concentrations), and repeated measures (Time).

the software used to perform the statistical analysis?  

Response: Statistical Analysis Systems SAS (SAS Institute Inc., Cary, NC).

Results

Starting the presentation of the section with an acronym is just insane. Please, make a nice context mentioning why the methodologies chosen are important for the study... Saying that something was higher or lower is just describing what we can see in tables and figures. Science is about the meaning of things, the whys...

Q1: What were the criteria to define the ranging concentrations of Carao powder in yoghurts? 

Response: The concentrations were chosen because at 5.3 g/L where yogurt physical characteristics changes. Meaning, that at this concentration the products defects can be evidently detected.

Lines 336-339: Please, redact this part with some further context.

Response: Lines 336-339 were redact with more context.

Line 347: A sentence with 5 words and 2 references?! Please, make an effort to improve this discussion.

Response: The discussion was improved.

Q2: Why the authors do not characterize the Carao extract by HPLC?

Response: Carao was not characterized due to lack of resources

Reviewer 2 Report

Article is well written however, modification/ improvement are required.

1. Hardness and rheology of yogurt (control and fruit powder added yogurt) are required.

2. Is it a stirrer type or set type yogurt ?

3. 3 ml of culture was mixed with how much mixture of milk ? what was initial CFU ? How inoculum was prepared ? Why inoculum was added at 85 degree C ? It is not acceptable. Starter culture can survive upto 50 degree C.

4. What was formula to measure inhibition of lipase activity ?

5. What were polyphenol and flavonoid content in freeze dried Carao (C. grandis) powder ? what is polyphenol and flavonoid profile ?

6. Microbiology analysis (line 103-110) is not appropriate, when the mixed culture was used, MRS cannot differentiate.

7. Why "enzyme activity" in axis of Figure 2 ? Is it inhibitory activity ? 

8. Antioxidants (polyphenols and flavonoids) profiles in yogurts are required.

9. Quality of pictures are needed to improved, specially Figure 2, Figure 9 and Figure 11.

10. Results and discussion sections are needed to improve. Limitations of investigation and further scope of investigation are needed to incorporate. 

11. Reference styles are not maintained in proper way. Sometimes name citation and sometimes number citations are noted. 

Language is readable. 

Author Response

. Hardness and rheology of yogurt (control and fruit powder added yogurt) are required.

Response: These analyses were not performed since the main objective of this paper was to study the influence of carao fortification into yogurt in functional properties. In our case was to study intestinal barriers disorder in caco2 cells and antidiabetic potential in rats’ studies.

  1. Is it a stirrer type or set type yogurt ?

Response: Set type of yogurt.

  1. 3 ml of culture was mixed with how much mixture of milk ?

Response: 1 ml per gallon, and 3 ml per 3 gallons of each culture.

what was initial CFU ?

Response: 108 per mL of culture

How inoculum was prepared ?

Response: The culture was purchased not prepared

Why inoculum was added at 85 degree C ? It is not acceptable. Starter culture can survive up to 50 degree C.

Response: The inoculum was added at 41 degree C. Milk was pasteurized first at 85 degree C. the sentence was rephrased and a reference was added to clarify the yogurt process.

“Aleman, R. S., Cedillos, R., Page, R., Olson, D., & Aryana, K. (2023). Physico-chemical, microbiological, and sensory charac-teristics of yogurt as affected by ingredients that help treat leaky gut. Journal of Dairy Science”

  1. What was formula to measure inhibition of lipase activity ?

Response: I = 100 – (AS/AC) *100                               

Whereas is the difference between the absorbance of the sample and the absorbance of the blank and AC is the difference between the absorbance of the control and the absorbance of the blank.

  1. What were polyphenol and flavonoid content in freeze dried Carao (C. grandis) powder ? what is polyphenol and flavonoid profile ?

Response: These analyses were not done since Fuentes et al., 2021 did the characterization.

Fuentes, J. A. M., López-Salas, L., Borrás-Linares, I., Navarro-Alarcón, M., Segura-Carretero, A., & Lozano-Sánchez, J. (2021). Development of an innovative pressurized liquid extraction procedure by response surface methodology to recover bioactive compounds from carao Tree Seeds. Foods, 10(2), 398.

  1. Microbiology analysis (line 103-110) is not appropriate, when the mixed culture was used, MRS cannot differentiate.

Response: I apologize for the lack of clarity.

  1. thermophilus was enumerated by using S. thermophilus (ST) agar, which has sucrose, K2HPO4, Bacto yeast extract, Bacto Tryptone and agar , while and L. bulgaricus (LB) was enumerated by using Lactobacillus MRS agar (Formulated with MRS broth powder and agar).

The microbial analysis were done as Alina and Aryana, 2021.

Buchilina, A., & Aryana, K. (2021). Physicochemical and microbiological characteristics of camel milk yogurt as influenced by monk fruit sweetener. Journal of Dairy Science, 104(2), 1484-1493.

  1. Why "enzyme activity" in axis of Figure 2 ? Is it inhibitory activity ?

Response: enzyme activity was changed to enzymatic inhibitory activity

  1. Antioxidants (polyphenols and flavonoids) profiles in yogurts are required.

Response: These analyses were not done due to lack of resources HPLC analysis are needed for this evalautions.

  1. Quality of pictures are needed to improved, specially Figure 2, Figure 9 and Figure 11.

Response: the quality of the pictures were improved.

  1. Results and discussion sections are needed to improve. Limitations of investigation and further scope of investigation are needed to incorporate.

Response: the discussion is improved.

  1. Reference styles are not maintained in proper way. Sometimes name citation and sometimes number citations are noted.

Response: The reference format and style were improved.

Reviewer 3 Report

The manuscript of Aleman et al. is very promising, well structured, with many different methodologies applied and very interesting results. Discussion needs to be revised in a way that more similar studies are incorporated in order to compare results, as well as more hypotheses should be done on the potential action of Carao and its constituents according to the findings.

Some minor comments:

Abstract

-line 34. fed with

Introduction

-42. Since it leads

-65 medicinal use?

Materials and Methods

-154. Whereas what?

-line 160, please rephrase the sentence

-line 217 something I smissing

-line 239 slight modification of what?

-2.12. Animal models. It is not clear what exact animals you used, animals fed with yogurt or yogurt and Carao were diabetic? Please review the paragraph in order to be clearer to the reader what animals models you had, maybe list them separately with detailed description.

-line 264. t?

-Line 265 please remove and

-line 286 posthoc?

Results

Figures. The specific use of a, b, c in figures of this manuscript is not very clear. For example, if different letters represent significant differences among control and (CY) samples what does b and c represent in Fig 3a? Please reconsider showing statistically significant differences in a more easily readable way for the authors. Also, please correct inflamatory to inflammatory in all figures.

-314. Demonstrating

-line 316. Please rephrase the sentence, the meaning is not clear

-318, 325. Since you mention not statistical significance how do you refer to higher values, please explain.

-333 bacteria cultures?

-336 has been shown

-334. something is missing from the sentence

-Table 1. Level or concentration?

-359. Diabetic problems> maybe diabetic complications?

-379. 394 figure 2a/2b?

-381 please rphrase

-395 indicating

-403 hypothesizing?

-394 if it is significant please show it in the aforementioned figure

-414 the cells or in cells?

-3.6 why you did not used genistein in this experiment? If you had please describe it clearer

-line 441, again not clear which are different according to text and figure

-445 and figure 9 misplace between 9A and B are zo-1 and occluding or claudin, please see text and figure

-510 compared to

- fed with

-558 oral glucose differences are statistically significant, please indicate in fig

-584 please rephrase

Conclusion

It is not clear whether you have performed this study in order to show potential activities of Carao on diabetes or on diseases related to Intestinal barrier dysfunction such as inflammatory bowel disease. Introduction focus on diabetes but conclusion mentions other diseases as well. If your aim was to show a potential action mainly on intestinal barrier in general and then on diabetes, you should reorganize your introduction and maybe add more on the significance of intestinal barrier in total and then in diabetes. If your aim was only diabetes then you should be more focused on that throughout the text.

Minor english corrections are needed

Author Response

comments and Suggestions for Authors

The manuscript of Aleman et al. is very promising, well structured, with many different methodologies applied and very interesting results. Discussion needs to be revised in a way that more similar studies are incorporated in order to compare results, as well as more hypotheses should be done on the potential action of Carao and its constituents according to the findings.

Some minor comments:

Abstract

-line 34. fed with

Response: “fed with” was placed…

Introduction

-42. Since it leads

Response: “Since it leads” was placed…

-65 medicinal use?

Response: medicinal use was deleted…

Materials and Methods

-154. Whereas what?

Response: Whereas was changed to Where AS

-line 160, please rephrase the sentence

Response: The suggested sentence was rephased

-line 217 something I smissing

Response: the sentence suggested was modified was recommended…

-line 239 slight modification of what?

Response: The expression of “slight modification” was deleted.

-2.12. Animal models. It is not clear what exact animals you used, animals fed with yogurt or yogurt and Carao were diabetic? Please review the paragraph in order to be clearer to the reader what animals models you had, maybe list them separately with detailed description.

Response: The rats models were listed were roman numbers

-line 264. t?

Response: T in the sentence was deleted

-Line 265 please remove and

Response: “and” was removed…

-line 286 posthoc?

Response: postdoc was changed to post hoc

Results

Figures. The specific use of a, b, c in figures of this manuscript is not very clear. For example, if different letters represent significant differences among control and (CY) samples what does b and c represent in Fig 3a? Please reconsider showing statistically significant differences in a more easily readable way for the authors. Also, please correct inflamatory to inflammatory in all figures.

Response: inflamatory was corrected to inflammatory. In Fig 3a, the keynote was adjusted to make easy readable the statistically significant among treatments.

-314. Demonstrating

Response: demonstrate was changed to demonstrating

-line 316. Please rephrase the sentence, the meaning is not clear

Response: The sentence was rephrased as suggested…

-318, 325. Since you mention not statistical significance how do you refer to higher values, please explain.

Response: Lines -318-325 were restated.

-333 bacteria cultures? S. thermophilus and L. bulgaricus

Response: Lines -333 were rephased.

-336 has been shown

Response: has been shown was placed…

-334. something is missing from the sentence

Response: The sentence was rephased.

-Table 1. Level or concentration?

Response: the table 1 title was changed to “Effect of carao concentration on the Total phenolic content (TPC), total flavonoid content (TFC), and antioxidant activity of freeze-dried yogurt.”

-359. Diabetic problems> maybe diabetic complications?

Response: diabetic problems was changed to “diabetic complications”.

-379. 394 figure 2a/2b?

Response: figure 2a/2b are placed correctly.

-381 please rephrase.

Response: the sentence was rephrased.

-395 indicating

Response: indicated was changed to “indicating”.

-403 hypothesizing?

Response: hypothesizing was deleted.

-394 if it is significant please show it in the aforementioned figure

Response: Sentence in line 394 was rephased.

-414 the cells or in cells?

Response: the cells was changed to in cells

-3.6 why you did not used genistein in this experiment? If you had please describe it clearer

Response: The control of this experiment was healthy cells (Cells just treated with growth media). The primary goal of this experiment was to compare yogurt samples ((0, 1.3, 2.65, and 5.3 g/L).

-line 441, again not clear which are different according to text and figure

Response: the sentence was rephased as suggested.

-445 and figure 9 misplace between 9A and B are zo-1 and occluding or claudin, please see text and figure

Response: The line 445 was rephased as suggested.

-510 compared to

Response: compared to was included

- fed with

Response: fed with was incorporated

-558 oral glucose differences are statistically significant, please indicate in fig

Response: the sentence was rephrased…

-584 please rephrase

Response: the sentence was rephrased…

Conclusion

It is not clear whether you have performed this study in order to show potential activities of Carao on diabetes or on diseases related to Intestinal barrier dysfunction such as inflammatory bowel disease. Introduction focus on diabetes but conclusion mentions other diseases as well. If your aim was to show a potential action mainly on intestinal barrier in general and then on diabetes, you should reorganize your introduction and maybe add more on the significance of intestinal barrier in total and then in diabetes. If your aim was only diabetes then you should be more focused on that throughout the text.

Response: introduction was improved as suggested.

Round 2

Reviewer 2 Report

Responses from authors are appreciable. Quality of figures are not well. Depths of X axis and Y axis will be similar and black in all cases. Figure 2 need to be portrait (similar like FIG. 1 and FIG 3) not the landscape. In some figures, error bars are not understandable, example fig 9. In whole manuscript, g/L need to according to rule of journal. Subheadings are not homogeneous, example in Figure 11, those are a and b; where as, in Figure 9, they are A.) and B.). Correction are required.

English language is fine.

Author Response

Responses from authors are appreciable. Quality of figures are not well. Depths of X axis and Y axis will be similar and black in all cases. Figure 2 need to be portrait (similar like FIG. 1 and FIG 3) not the landscape. In some figures, error bars are not understandable, example fig 9. In whole manuscript, g/L need to according to rule of journal. Subheadings are not homogeneous, example in Figure 11, those are a and b; where as, in Figure 9, they are A.) and B.). Correction are required.

Response: The figures were improved as suggested….